# GAVEL: AGENT MEETS CHECKLIST FOR EVALUATING LLMS ON LONG-CONTEXT LEGAL SUMMARIZATION

## ABSTRACT

Large language models (LLMs) are increasingly applied in legal practice, with case summarization being a key long-context task where cases often exceed 100K tokens across multiple documents. Existing evaluation methods rely on checklist comparisons but use coarse-grained extraction that merges multiple values into single text blocks, missing partial matches when comparing them. They also overlook content beyond predefined checklist categories and lack writing style evaluation. In this paper, we introduce GAVEL-REF, a reference-based evaluation framework that improves checklist evaluation through multi-value extraction with supporting text, and further incorporates residual fact and writing-style assessments. Using GAVEL-REF, we move beyond the single aggregate scores reported in prior work to systematically evaluate 12 frontier LLMs on 100 legal cases ranging from 32K to 512K tokens, primarily from 2025. Our detailed analysis reveals Gemini 2.5 Pro, Claude Sonnet 4, and Gemini 2.5 Flash achieve the best performance (around 50 $S_{\text{GAVEL-REF}}$), showing the difficulty of the task. These top models show consistent patterns: they succeed on simple checklist items (e.g., filing date) but struggle on multi-value or rare ones such as settlements and monitor reports. As LLMs keep improving and may eventually surpass human summaries, we also explore checklist extraction directly from case documents. We experiment with three different methods: end-to-end with long-context LLM, chunk-by-chunk extraction, and our newly developed autonomous agent scaffold, GAVEL-AGENT. Our results show strong potential for the agent approach in long-context processing: compared to the best GPT-4.1 end-to-end setup, Gavel-Agent with Qwen3 reduces token usage by 36% while achieving competitive performance (only 7% lower in $S_{\text{checklist}}$). We will release our code and annotations publicly to facilitate future research on long-context legal summarization.

## 1 INTRODUCTION

Large language models (LLMs) (Brown et al., 2020; Achiam et al., 2023) are now widely adopted across various industries and professions. The legal sector has been particularly active (Frankenreiter & Nyarko, 2022; Ziffer, 2023), with startups such as Harvey building AI for lawyers. Among legal applications, court document summarization stands out as both practically important and technically challenging. A single litigation case can easily involve dozens of court documents, including complaints, orders, and rulings, with a combined length exceeding 100,000 tokens, roughly equivalent to 80 news articles or a 300-page novel. Unlike news summarization, where lead sentences often suffice (Narayan et al., 2018; Liu & Lapata, 2019), or fiction books, where events can be summarized sequentially (Chang et al., 2024), legal cases require tracking interconnected arguments across multiple documents. It requires maintaining exact chronology, preserving relationships between parties, claims, and rulings, and ensuring that cross-references between filings remain accurate. Moreover, a collection of expert-written case summaries is available (Shen et al., 2022) to serve as a gold standard for this task. The combination of these factors makes legal summarization an ideal testbed for assessing LLMs' long-context capabilities; meanwhile, it also calls for more reliable and comprehensive evaluation methodologies than those currently in use.

To evaluate summarization, researchers have moved beyond traditional n-gram metrics such as ROUGE (Lin, 2004) and BLEU (Papineni et al., 2002), developing checklist-based methods with LLM-as-judge (Min et al., 2023; Pereira et al., 2024; Lee et al., 2024; Lin et al., 2025). The most

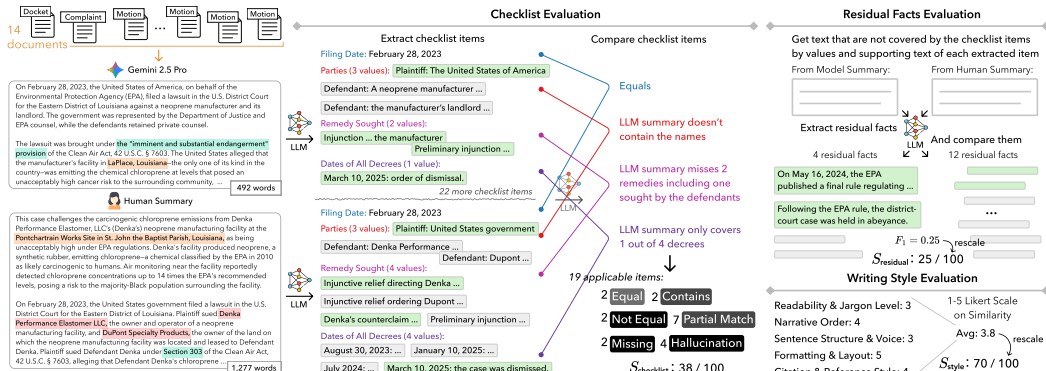

Figure 1: Example of evaluating a Gemini 2.5 Pro summary with GAVEL-REF, which contains: checklist evaluation supporting both string-wise and list-wise comparisons, residual fact evaluation, and writing-style evaluation. An interesting finding is that many modern LLMs tend to omits specific names of people or organizations—in this case, the defendant companies; and in other cases even the U.S. president's name. Light green background indicates matched values.

relevant recent work is ExpertLongBench (Ruan et al., 2025), which includes legal summarization in its benchmark. They ask legal experts to define 26 checklist items commonly found in legal case summaries (e.g., filing date, remedy sought, decrees), and an LLM is used to extract these items from both human- and model-generated summaries for item-by-item comparison. This marks an important step toward structured and interpretable evaluation, but the approach still has two key limitations: (i) many checklist items (e.g., remedy sought) may contain multiple distinct values (see Figure 1), yet existing method treats them as a single text block, making it difficult to capture partial matches. (ii) the evaluation is restricted to predefined checklist items, overlooking additional useful content outside the checklist and other qualities such as readability or formatting. Furthermore, ExpertLongBench and other existing benchmarks (Yen et al., 2024; Ruan et al., 2025) are built to evaluate LLMs across many tasks, with legal summarization as one of them. They provide valuable benchmarking of these models, but they naturally do not aim to offer detailed analysis of how modern LLMs perform on legal summarization specifically—for example, which checklist items models systematically struggle with or whether they capture non-checklist information that human experts often include. Finally, as LLMs continue to advance, they may surpass human-written summaries. This motivates deriving checklists directly from case documents to reduce reliance on human references while enabling test-time feedback. However, it is unclear from existing work whether current LLMs or agent-based methods can effectively handle this long-context extraction task.

In this paper, we address all three gaps. Firstly, we introduce GAVEL-REF (see Figure 1), which improves checklist evaluation by enabling list-wise comparison, and we further extend it with assessments of residual facts (information beyond the 26 checklist items) and writing style. We compare GAVEL-REF, with different LLMs as its backbone, against human annotators who perform the same task. Specifically, we collect 5,442 item-level annotations on 40 long summaries (averaging 1,130 words each), 450 checklist comparison judgments, and 375 style similarity ratings, totaling 150 hours of human effort. Our results show that GAVEL-REF using open-source GPT-oss 20B (Agarwal et al., 2025) and Qwen3 (Yang et al., 2025) models achieves performance comparable to GPT-5, demonstrating that large-scale automatic evaluation can be both reliable and cost-effective.

Secondly, using GAVEL-REF, we evaluate 12 LLMs, including proprietary models (GPT-5 and Gemini 2.5) and open-source models (GPT-oss and Qwen3), on 100 cases spanning 32K to 512K tokens, far beyond the 128K limit of prior work. To reduce data contamination, 83% of cases are new from 2025 and likely unseen by the models. Our main findings are: (i) Gemini 2.5 Pro, Claude Sonnet 4, and Gemini 2.5 Flash achieve the best summaries with $S_{\text{GAVEL-REF}}$ score of 50 out of 100, underscoring the difficulty of long-context legal summarization. (ii) Proprietary models outperform open-source ones at the 30B scale, with open-source models such as Gemma3 (Team et al., 2025) and Qwen3 degrading more drastically as case length increases. (iii) GPT-4.1 best captures residual facts while GPT-5 tends to produce checklist-like and verbose summaries even when prompted for narrative style, while Claude and Gemini models most closely match human style. (iv) Top models handle single-value items well, multi-value items less reliably, and struggle most with related cases and monitoring reports.

Thirdly, for extracting checklists directly from case documents, beyond standard approaches such as feeding all documents into a long-context LLM or chunking them and extracting items iteratively, we develop a novel agent scaffold, GAVEL-AGENT. It equips LLMs with six tools for autonomously navigating documents and locating checklist items, emulating how humans process case documents. Our experiments show that end-to-end extraction with GPT-4.1 achieves the best overall performance, with GAVEL-AGENT using Qwen3 performing very closely behind. The advantage of GAVEL-AGENT is efficiency: it uses 36% fewer tokens than the GPT-4.1 end-to-end setup and 59% fewer than the chunk-by-chunk approach, highlighting the strong potential of agents for long-context tasks. Compared to extracting from summaries, checklist extraction from full documents still lags significantly, pointing to future work on long-context LLMs and long-horizon agents.

In summary, our contributions are as follows:

1. We introduce GAVEL-REF, a reference-based evaluation framework for legal summarization that provides a comprehensive assessment via checklist, residual fact, and writing style evaluation.

2. Using GAVEL-REF, we systematically evaluate 12 frontier LLMs across different case lengths and reveal their gaps in capturing complex legal checklist items with a detailed analysis.

3. We explore checklist extraction from case documents using three different approaches: end-to-end, chunk-by-chunk, and GAVEL-AGENT—our autonomous agent scaffold.

## 2 GAVEL-REF—A REFERENCE-BASED EVALUATION FRAMEWORK

We introduce GAVEL-REF (Fig. 1), an automatic, reference-based evaluation framework for legal summarization with three complementary components. First, *checklist evaluation* extracts values and supporting text for 26 items(e.g., filing date, parties, decrees). Second, *residual facts evaluation* captures and scores content beyond the checklist. Third, *writing style evaluation* compares model summaries' similarity to human references across five aspects. Prompts are in App. G.

### 2.1 METHOD DESCRIPTION

**Checklist Evaluation.** ExpertLongBench (Ruan et al., 2025) presents a checklist-based evaluation framework for long-form generation, where legal experts create a checklist of 26 key items for legal summaries. For each item $c_i$, an LLM extracts the corresponding information $H(c_i)$ from the model summary and $R(c_i)$ from the reference, then determines containment relationships between them. While this provides a solid foundation, we identify limitations and improve it as follows:

*Improvement 1: Multi-value extraction with supporting text.* We find that checklist items contain multiple values 76% of the time (e.g., several filings or factual bases in a case). However, prior method extracts all information as a single text block and performs a binary comparison. This misses partial overlaps—for example, five filings vs. five different filings with three overlaps is scored the same as a total mismatch.

To address this limitation, we restructure extraction so that each checklist item $c_i$ yields a list of values with supporting text: $H(c_i) = \{(v_{i,1}, s_{i,1}), (v_{i,2}, s_{i,2}), \ldots, (v_{i,n}, s_{i,n})\}$, where $v_{i,j}$ is the $j$-th extracted value for checklist item $c_i$, and $s_{i,j}$ is a set of verbatim snippets grounding it. Supporting text not only justifies values but also helps us later identify residual facts that fall outside the checklist. For comparison, single-value items are judged by an LLM as equal, A contains B, B contains A, or different, while multi-value items use element-wise matching to identify overlaps and uniques.

*Improvement 2: Score aggregation.* When some checklist item doesn't exist in the case documents, both the model and human naturally won't include it in their summaries. However, the original method counts it as a correct match. This inflates the denominator and reduces the penalty for actual errors. As non-applicable items dilute the score calculation, errors like hallucinations or omissions of key items have less impact on the final score.

To address this issue, we compute scores based only on applicable items, defined as those present in at least one summary. The final score is: $S_{\text{checklist}} = \frac{100}{|A|} \sum_{c_i \in A} m_i$, where $A$ is the set of applicable

checklist items, and the matching score $m_i$ is defined as:

$$m_i = \begin{cases} \begin{cases} 1 & \text{if } H(c_i) = R(c_i) \\ 0.5 & \text{if } H(c_i) \subset R(c_i) \text{ or } H(c_i) \supset R(c_i) \\ 0 & \text{otherwise} \end{cases} & \text{if single-value} \\ F_1(H(c_i), R(c_i)) & \text{if multi-value} \end{cases} \tag{1}$$

For single-value items, we assign full points for equality, half points for containment, and zero otherwise. For multi-value items, we use $F_1$ as the matching score.

**Residual Facts Evaluation.** While the checklist captures essential case information, summaries sometimes include details beyond these 26 items. To evaluate this additional content, we first identify text segments not covered by the checklist. We use two-stage matching to precisely identify uncovered text: first against the extracted values alone, then against their supporting sentences if unmatched. This prevents over-coverage—such as when a filing date's support text also contains other legal facts. We then use an LLM to extract atomic facts (termed "residual facts") from these uncovered segments and evaluate them using the same list-wise comparison method as in our checklist evaluation. The resulting $F_1$ score (scaled to 0-100) is the $S_{\text{residual}}$.

**Writing Style Evaluation.** Beyond content, we measure how closely model summaries match human ones in writing style. We emphasize similarity over quality, as quality is subjective (e.g., preference for narratives vs. bullet points). Five aspects are rated on a 1–5 Likert scale (1 = completely different, 5 = identical): Readability & Jargon Level, Narrative Order, Sentence Structure & Voice, Formatting & Layout, Citation & Reference Style. We average these scores, subtract 1, and multiply by 25 to obtain $S_{\text{style}}$ on a 0-100 scale. See Appendix C for definitions of each aspect.

## 2.2 THE OVERALL GAVEL-REF SCORE

To combine all three components into a final score for benchmarking LLMs or use as a reward signal, we compute a weighted linear combination:

$$S_{\text{GAVEL-REF}} = (1 - r) \cdot \alpha \cdot S_{\text{checklist}} + r \cdot \alpha \cdot S_{\text{residual}} + (1 - \alpha) \cdot S_{\text{style}} \tag{2}$$

where $\alpha$ controls the balance between content and style, and $r$ is the proportion of residual content in the reference summary (total residual text spans length divided by summary length). This dynamically weights $S_{\text{checklist}}$ and $S_{\text{residual}}$ based on their relative importance in each summary—more residual content increases the weight on $S_{\text{residual}}$. We set $\alpha$ as 0.9 throughout our paper.

## 2.3 META-EVALUATION OF GAVEL-REF

To validate that GAVEL-REF accurately captures summary quality, we recruit four in-house annotators to perform the same evaluation tasks as the LLM—extracting checklist items, comparing checklist item values, and rating writing style similarity—then measure the agreement between LLM and human annotations.

**Collecting Human Annotations.** To evaluate LLMs' ability to *extract checklist items*, we annotated 40 long case summaries (avg. 1,130 words) to stress-test the models: if the LLM can accurately extract checklist items from these longer summaries, it should perform at least as well on the shorter ones used in the main model evaluation. Since extracting all 26 checklist items from scratch is time-consuming, annotators start from GPT-5's extractions. Using our paragraph-by-paragraph review interface modified from Thresh (Heineman et al., 2023), annotators add missing values, correct extractions and supporting text, or delete incorrect values. Each summary annotation takes approximately one hour. Figures 13 to 22 in the Appendix show an example of our annotations on a case summary, covering all 26 checklist items. In total, we collect 70 summary-level annotations covering 5,442 item-level annotations, where the ten longest summaries (averaging 1,695 words) receive triple annotations, with adjudication by a fourth annotator. The remaining 30 summaries receive single annotations. To evaluate LLMs' ability to *compare checklist values*, annotators assess 150

item pairs from model and reference summaries (100 multi-value, 50 single-value), drawn from diverse LLMs for generalizability. For single-value pairs, they perform 4-class classification: equal, A contains B, B contains A, or different. For multi-value pairs, they match elements from list A to list B. Annotations are aggregated by majority vote: for single-value items, we take the class with $\geq$ two votes (no cases had all three labels differ); for multi-value items, we keep matches identified by $\geq$ two annotators. To evaluate LLM's ability to *rate writing style similarity*, we annotate 25 model-reference summary pairs. Annotators rate similarity across five style aspects using 1-5 Likert scales, with three annotations per pair. Final scores are the median across annotators.

All annotators are paid $18 USD per hour, with a total cost of $3K USD. Appendix D provides training details, inter-annotator agreement results, and screenshots of the annotation interfaces.

**Metrics.** For *checklist comparison*, we use accuracy for single-value items (4-class classification) and matching-pairs F1 for multi-value items, which measures how accurately the LLM identifies correct matches between two lists. The best comparison model is then used to evaluate *checklist extraction*, computing $S$checklist against human-extracted checklist from the same summary. We also compute word-level coverage agreement on supporting text, measuring how often model and human agree on whether words are covered by checklist items or are residual. For *writing style rating*, we report Cohen's Kappa for LLM-human agreement.

**Results.** We select models based on two criteria: state-of-the-art performance and open-source availability. We prioritize open-source models for cost-efficient large-scale evaluation in Section 3. We evaluate five LLMs: GPT-5 and four open-source models—Qwen3 32B, Qwen3 30B-A3B, GPT-oss 20B, and Gemma3 27B. Table 1 presents the results. GPT-5 performs best at checklist extraction, with GPT-oss 20B second overall and showing much higher coverage than the other open-source models. Reasoning models perform better than Gemma3 27B on this task. However, Gemma3 27B outperforms all reasoning models on single string comparison and achieves comparable

| Model | Checklist Extraction | | Checklist Comparison | | Style |
| --- | --- | --- | --- | --- | --- |
| | $S_{\text{checklist}}$ | Coverage | Single | Multi | Rating |
| GPT-5 | **68.2** | **92.9%** | 0.567 | *0.847* | *0.115* |
| GPT-oss 20B | 64.4 | *83.7%* | 0.567 | 0.801 | **0.157** |
| Gemma3 27B | 54.1 | 75.3 % | **0.740** | 0.841 | 0.091 |
| Qwen3 32B | *65.5* | 66.0% | 0.600 | 0.820 | 0.084 |
| Qwen3 30B-A3B | 63.3 | 63.0% | *0.700* | **0.854** | -0.011 |

Table 1: Meta-evaluation results of five models in GAVEL-REF: Checklist Extraction ($S_{\text{checklist}}$ and word-level coverage agreement), Checklist Comparison (accuracy for single-value, matching $F_1$ for multi-value), and Writing Style Rating (Cohen's $\kappa$). **Bold**: best, *italic*: second best.

performance on list-wise comparison. GPT-oss 20B achieves the best alignment with human ratings of writing style. Based on these results, we use GPT-oss 20B for checklist extraction and style rating, and Gemma3 27B for checklist comparison in Section 3 when evaluating LLM summaries.

# 3 EVALUATION OF LLM LEGAL SUMMARIZATION WITH GAVEL-REF

Prior work (Yen et al., 2024; Ruan et al., 2025) have evaluated LLM legal summarization on legal cases up to 128K that are before 2024. As the latest LLMs now handle 1M tokens and have pre-trained knowledge up to 2025, in this work, we want to shed light on how these modern models perform on much longer context using 2025 legal cases beyond their training cutoffs. With GAVEL-REF, we evaluate 12 LLMs that span both proprietary and open-source models across 5 different case length scales: 32K, 64K, 128K, 256K, 512K tokens (measured by the GPT-4o tokenizer). For each scale, we select 20 cases whose token counts fall within $\pm 20\%$ of the target length. Of the 100 cases, 83 are filed in 2025 (using the filing date of the first docket entry). The remaining 17 cases (14 in the 512K bin and 3 in the 32K bin) are from earlier years due to limited availability—especially for the 512K bin. At the time of writing (7-8 months into 2025), very few cases have accumulated enough documents to reach 512K tokens; on average, cases in this bin take about 1.5 years to reach that length. Since the models have varying context limits and some cases exceed these limits, we truncate by proportionally removing tokens from the end of each document, following prior work.

| Model | Overall Evaluation: $S_{\text{GAVEL-REF}}$ | | | | | | Checklist Evaluation: $S_{\text{checklist}}$ | | | | | | Residual Facts Evaluation: $S_{\text{residual}}$ | | | | | | Writing Style Evaluation: $S_{\text{style}}$ | | | | | |
|---|---|---|---|---|---|---|---|---|---|---|---|---|---|---|---|---|---|---|---|---|---|---|---|---|
| | 32K | 64K | 128K | 256K | 512K | all | 32K | 64K | 128K | 256K | 512K | all | 32K | 64K | 128K | 256K | 512K | all | 32K | 64K | 128K | 256K | 512K | all |
| **Proprietary** | | | | | | | | | | | | | | | | | | | | | | | | |
| Gemini 2.5 Pro (1M) | **54.0** | 49.2 | 53.2 | **49.1** | **49.3** | **51.0** | 54.2 | **53.8** | 55.7 | **53.0** | 51.9 | **53.7** | 1.8 | 6.5 | 5.4 | 12.1 | 7.9 | 7.2 | **74.5** | 70.0 | 72.5 | 70.5 | 67.5 | 71.0 |
| Claude Sonnet 4 (200K) | 52.3 | 50.3 | 51.5 | 48.2 | 48.5 | 50.1 | 51.4 | 52.9 | 53.6 | 52.4 | 50.3 | 52.1 | **8.5** | 20.6 | 5.2 | 7.0 | 7.9 | 9.8 | 72.0 | 71.5 | **76.2** | 70.0 | 65.5 | **71.0** |
| Gemini 2.5 Flash (1M) | 50.9 | 48.4 | **53.9** | 47.3 | 49.3 | 50.0 | 51.7 | 51.5 | 55.5 | 51.1 | **52.1** | 52.4 | 3.8 | 9.1 | 13.5 | 8.4 | 12.1 | 9.6 | 65.0 | 69.5 | 72.2 | **71.2** | **69.2** | 69.5 |
| Claude Opus 4.1 (200K) | 51.9 | 49.8 | 51.6 | 47.7 | 47.7 | 49.7 | 51.9 | 52.0 | 52.1 | 51.3 | 49.6 | 51.4 | 5.2 | 13.0 | 15.9 | 9.0 | 6.0 | 9.9 | 70.8 | **72.5** | 75.2 | 69.2 | 67.2 | 71.0 |
| GPT-4.1 (1M) | 51.6 | **50.4** | 51.7 | 47.0 | 44.0 | 49.0 | 50.6 | 52.8 | 51.5 | 48.6 | 44.9 | 49.7 | 8.4 | 18.3 | **22.8** | **22.3** | **13.0** | **17.2** | 69.0 | 72.2 | 71.5 | 68.0 | 60.8 | 68.3 |
| GPT-5 (400K) | 48.6 | 48.7 | 48.6 | 48.7 | 47.8 | 48.5 | 50.0 | 50.9 | 50.3 | 51.6 | 49.4 | 50.4 | 7.5 | **21.9** | 16.0 | 16.1 | 11.3 | 14.6 | 50.0 | 53.8 | 61.0 | 63.8 | 67.2 | 59.1 |
| **Open-source** | | | | | | | | | | | | | | | | | | | | | | | | |
| GPT-oss 20B (128K) | 49.0 | 47.0 | 47.3 | 43.5 | 42.5 | 45.9 | 49.1 | 50.5 | 49.8 | 47.2 | 43.8 | 48.1 | 2.8 | 9.8 | 2.6 | 7.3 | 10.3 | 6.7 | 67.2 | 69.8 | 70.5 | 60.8 | 59.2 | 65.5 |
| Qwen3 32B (131K) | 48.4 | 48.1 | 46.8 | 42.3 | 38.6 | 44.8 | 48.5 | 51.0 | 48.1 | 45.7 | 40.5 | 46.8 | 0.0 | 14.8 | 7.8 | 6.2 | 3.1 | 6.6 | 69.8 | 68.0 | 71.5 | 64.2 | 57.5 | 66.2 |
| Qwen3 14B (131K) | 50.7 | 43.1 | 42.4 | 41.5 | 38.3 | 43.2 | 50.9 | 45.6 | 43.4 | 44.2 | 40.0 | 44.8 | 7.1 | 6.0 | 2.6 | 4.8 | 6.1 | 5.2 | 71.0 | 69.0 | 70.2 | 65.2 | 57.5 | 66.6 |
| Qwen3 30B-A3B (262K) | 49.9 | 43.2 | 41.7 | 33.8 | 33.6 | 40.4 | 50.6 | 47.1 | 43.8 | 34.0 | 33.9 | 41.9 | 0.0 | 0.0 | 2.5 | 3.9 | 5.0 | 2.5 | 66.0 | 64.0 | 64.0 | 61.5 | 55.5 | 62.2 |
| Gemma3 12B (128K) | 46.1 | 41.1 | 40.9 | 32.7 | 28.4 | 37.8 | 45.3 | 42.7 | 41.6 | 35.1 | 29.0 | 38.7 | 7.6 | 6.4 | 8.7 | 1.5 | 3.9 | 5.4 | 70.8 | 63.5 | 62.0 | 55.5 | 47.5 | 59.9 |
| Gemma3 27B (128K) | 44.4 | 39.0 | 34.8 | 31.2 | 30.4 | 35.9 | 43.9 | 41.4 | 35.3 | 33.0 | 31.1 | 36.9 | 2.6 | 4.3 | 8.4 | 0.0 | 2.0 | 3.4 | 68.0 | 62.0 | 63.2 | 57.8 | 48.5 | 59.9 |

Figure 2: Benchmarking results of 12 LLMs on long-context legal summarization with our GAVEL-REF framework across case lengths from 32K to 512K tokens. Models are ordered by $S_{\text{GAVEL-REF}}$ on all cases. Gemini 2.5 Pro leads, with all top six positions held by proprietary models.

## 3.1 BENCHMARKING RESULTS FOR 12 MODELS

Figure 2 shows GAVEL-REF evaluation results for 12 models across different case length bins. Figure 6 in the Appendix additionally shows the summary length of each model in each length bin, compared to human summary length.

**Gemini 2.5 Pro, Claude Sonnet 4, and Gemini 2.5 Flash are the top three models.** Proprietary models consistently outperform open-source ones by a clear margin. Overall, Gemini 2.5 Pro achieves the best performance with an $S_{\text{GAVEL-REF}}$ of 51.0, while the best open-source model, GPT-oss 20B, reaches 45.9. Interestingly, GPT-5 is the weakest among the proprietary models, largely due to its overly verbose summaries, which we analyze in more detail in the paragraphs below. Within the Claude family, Sonnet 4 slightly outperforms Opus 4.1. To understand which checklist items drive this gap, we present checklist item–level performance for each LLM in Figures 10–12 in the Appendix. We find that Sonnet 4 is stronger in identifying items such as Cause of action, Class action vs. individual, and Remedy sought than Opus 4.1.

**All models degrade as case length increases, with larger drops for open-source models.** We observe a consistent pattern: $S_{\text{GAVEL-REF}}$ decreases as case length grows, and models perform worst on the 256K and 512K bins. Even though models like Gemini 2.5 Pro, Gemini 2.5 Flash, and GPT-4.1 support a 1M-token context window, they still show noticeable drops on long cases—for example, Gemini 2.5 Pro is 4.7 points lower on 512K than on 32K cases, and GPT-4.1 drops by 7.6 points. Open-source models degrade even more on 256K and 512K cases, which is expected since they do not support such long contexts, and truncation of the case documents causes substantial information loss. These results call for scaffolded agents for long-context legal summarization.

**GPT-4.1 performs best on residual facts evaluation, with GPT-5 close behind.** Both models tend to capture more non-checklist details than other models. On average, the residual ratio $r$ (the proportion of residual content in the whole summary, Eq. 2) is 18.7% for GPT-4.1 and 18.4% for GPT-5. These are the only two models that exceed the human residual ratio of 11.1%; the next highest model, Claude Sonnet 4, is only 7.3%. As a result, GPT-4.1 and GPT-5 obtain the highest $S_{\text{residual}}$ of 17.2 and 14.6, respectively. However, these values are still below 20, indicating that the overlap between human residual facts and the residual facts captured by the models remains limited.

**Surprisingly, GPT-5 has the lowest writing-style rating, while Gemini and Claude models have the most human-like style.** Claude Opus 4.1, Sonnet 4, and Gemini 2.5 Pro all achieve $S_{\text{style}}$ of 71.0, whereas GPT-5 scores lowest at 59.1. As illustrated in Figure 9, GPT-5 often ignores the instruction to write in narrative form, instead producing sectioned summaries organized by checklist items, and tends to be very verbose—sometimes close to 1,000 words when the corresponding human summary is around 700 words. All models perform best on 64K–128K cases in terms of style similarity. On longer cases (256K–512K), every model's writing becomes less human-like, with similar drops across the board. From Figure 6, we see that in the 256K and 512K bins human summaries are

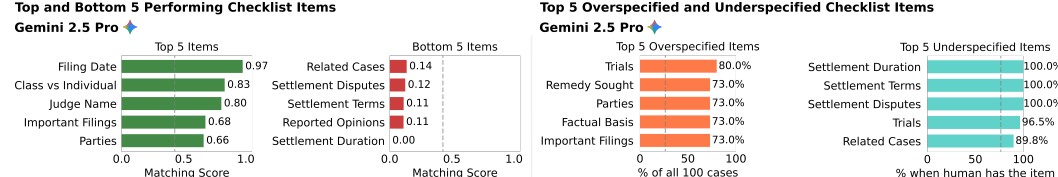

Figure 4: Gemini 2.5 Flash performance breakdown: top/bottom 5 checklist items by matching score and most frequently over/under-specified items. Overspecification measured as frequency across all 100 cases; underspecification as frequency among cases where human summary includes that item. Dashed lines are medians: 0.49 matching score, 59% overspecification, 70% underspecification.

around 1,200 words, while proprietary models (excluding GPT-5) typically produce summaries of 500–800 words. Open-source models are even more concise, usually under 400 words, and weaker models such as Gemma often stay below 300 words across all length bins.

### 3.2 How Top Models Handle Different Checklist Information

Figure 3 shows performance of the top five models across nine checklist groups, using the matching score $m_i$ (Eq. 1). All models follow a similar pattern. **They are good at extracting basic case information, legal foundations, and judge details**, scoring above 0.6. This makes sense as these groups contain mostly single-value items like filing date, cause of action, type of counsel, and judge name. **Performance drops noticeably for multi-value items.** Court rulings, decrees, and factual basis (context) prove more challenging, with scores around 0.4-0.5. Models must track multiple related pieces of information scattered across lengthy documents and determine which ones are important enough to include. **The models struggle most with related cases and settlements,** scoring below 0.2. The items in these groups appear rarely in the cases.

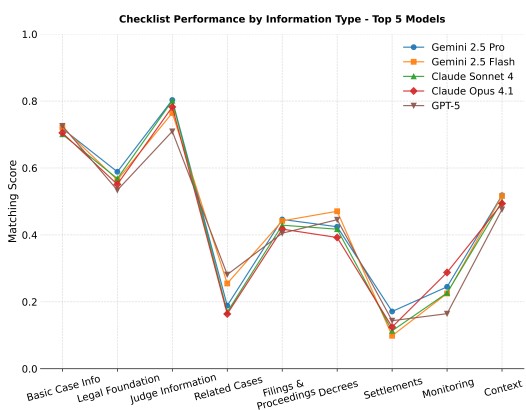

Figure 3: Top-5 LLMs' performance across checklist groups, struggling the most on rare items such as related cases and settlements.

### 3.3 Dissecting the Top Performer: Item-Level Analysis

Figure 4 analyzes Gemini 2.5 Pro's item-level performance, showing its top and bottom 5 checklist items plus consistently over- and under-specified items (see Appendix Figure 7 for top-3 models).

**Single-value items are Gemini's strength, while settlement details are its blind spots.** Filing date leads with a near-perfect matching score of 0.97, followed by other straightforward items such as Class action vs. Individual (0.83) and Judge name (0.80). For the next-best items, Important Filings and Parties, the scores fall below 0.7, and the median matching score across all 26 items is 0.43. In contrast, Gemini struggles dramatically with settlement-related information—scoring just 0.12, 0.11, and 0.00 on the three settlement items—while Related Cases and Reported Opinions are also among the weakest-performing items.

**Gemini 2.5 Flash tends to overspecify and underspecify checklist items with multiple values in its summaries.** All of the top five over-specified and under-specified items are multi-value items, with Trials appearing in both lists. This suggests that when multiple values are possible, the model has difficulty matching human judgments about which details to include. Settlement Duration, Settlement Terms, and Settlement Disputes are under-specified 100% of the time. Overall, the model is much more prone to under-specification than over-specification: the median overspecification rate is 26.5%, whereas the median underspecification rate is 76.5%.

# 4 EXTRACTING CHECKLIST FROM CASE DOCUMENTS

While reference-based evaluation effectively benchmarks summarization models, it requires hours of legal expert time per case to create human summaries, which cannot serve as a long-term gold standard once LLMs begin to surpass humans. Directly extracting checklists from case documents removes this dependency, enabling scalable evaluation, testing of superhuman models, and grounded suggestions during inference. To this end, we experiment with three methods: end-to-end extraction with long-context LLMs, processing the case documents chunk by chunk, and GAVEL-AGENT—an autonomous agent framework we develop to test whether LLMs can efficiently extract information by strategically searching and skimming rather than reading every word.

## 4.1 METHODS

**End-to-end.** We concatenate all case documents in chronological order and feed them to long-context LLMs. Instead of extracting all 26 checklist items at once, we query each item individually, which gives more accurate results.

**Chunk-by-chunk.** We split each document into 16K-token chunks, long enough to capture most documents while fitting within modern LLM context windows (32K+). At each step, the model receives the chunk text and current checklist state, then outputs an updated state—retaining existing values or adding new ones. Like end-to-end, we process documents chronologically and extract all 26 items. This mirrors multi-agent long-context methods (Zhang et al., 2024; Zhao et al., 2024), which segment text and process chunks independently.

**GAVEL-AGENT.** Unlike end-to-end or chunk-by-chunk methods that make models to read everything, human experts strategically search and skim for relevant information. To mimic this, we develop GAVEL-AGENT, an agent scaffold that lets LLMs navigate documents and extract checklist items autonomously. GAVEL-AGENT provides the LLM with six tools such as read a document, run regex searches across documents, and update checklist items. At each step, the model chooses a tool or issues a stop action based on the current state and history. Standard scaffolds append each tool call and response to agent's context. While working for short tasks, this approach breaks down in long cases (256K+ tokens, 50+ calls), where the context quickly balloons and the model must track information across an increasingly unwieldy history. Instead, GAVEL-AGENT refreshes the state after each tool call, giving LLM a clean snapshot including documents explored state, recent action details, etc. GAVEL-AGENT is fully customizable: users can define any checklist items, making it easy to transfer to domains like biomedical or financial extraction.

*Tools.* The following are the definitions of the six tools in GAVEL-AGENT:

- `list_documents()`: Returns all available documents with their metadata such as document type and token count. It is used to provide an initial catalog of the case.
- `read_document(doc_name, start_token, end_token)`: Reads a specific token range from a document, with a maximum of 10,000 tokens per call.
- `search_document_regex(pattern, doc_name/doc_names, top_k, context_tokens)`: Searches one, multiple or all documents using regex patterns, returning the top-k matches with surrounding context (100-1000 tokens).
- `get_checklist(item/items)`: Retrieves extracted values for specified checklist items.
- `append_checklist(patch)`: Adds new values for specific checklist items, supporting multiple values per item with required evidence (verbatim text, source document, and location).
- `update_checklist(patch)`: Replaces all values for specified checklist items, used for corrections or marking items as "Not Applicable" when no relevant information exists.

Both `append_checklist` and `update_checklist` use a `patch` structure that supports batch operations. Each patch contains an array of checklist keys to update, where each key maps to an array of extracted values, and every value includes (1) the value itself and (2) an array of supporting evidence (verbatim text, source document, and location). This structure ensures traceability from extracted information back to source documents.

*Context Management.* At each step, the LLM is given a system prompt high-level task instruction and tool descriptions, and a user prompt that contains user instruction (e.g., "Extract all 26 checklist

items"), the checklist definitions of the items to extract, a document catalog showing which parts have been explored, a summary of what has been extracted so far, and the recent action history. For action history, we maintain up to 100 tool calls: the five most recent include full responses (e.g., full text from `read_document`), while the other 95 are compressed to the tool name and brief outcome (e.g., "read 3,000 tokens", "updated filing date"). This gives the model enough awareness to avoid repeating actions while keeping the prompt compact.

### 4.2 IMPLEMENTATION DETAILS

**Model Selection.** For end-to-end extraction, we use GPT-4.1 with its 1M-token context. For chunk-by-chunk extraction, we test three open-source reasoning models: GPT-oss 20B, Qwen3 32B, and Qwen3 30B-A3B. For GavelAgent, we use Qwen3 30B-A3B and GPT-oss 20B, as both support 128K+ context natively, sufficient for context management.

**GAVEL-AGENT Configurations.** It is unclear whether agents perform better extracting multiple checklist items together—potentially using each document read more efficiently—or focusing on single items for higher accuracy. To study this trade-off, we test three setups: (1) one agent extracting all 26 items; (2) 9 agents for grouped items (e.g., filing date, parties, and counsel under "Basic Case Information"); (3) 26 agents, each handling a single item. See App. B for full checklist definitions.

### 4.3 META-EVALUATION

Following the evaluation of GAVEL-REF in Section 2.3, we evaluate extraction quality on 20 long cases. We use Gemma3 27B to compare each method's extracted checklist against the human-created checklist from the summary, computing the $S_{checklist}$ score. We also measure token usage (input and output) as efficiency.

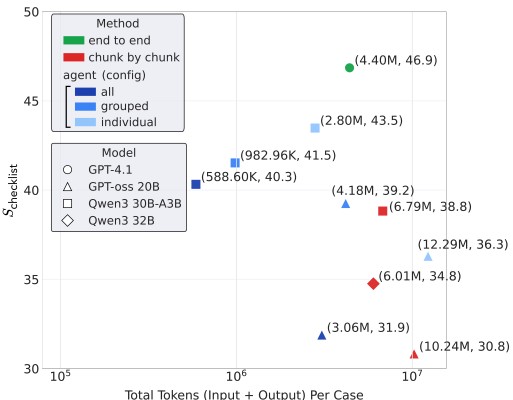

Figure 5: $S_{checklist}$ versus total token usage for different methods extracting from case documents.

**Results.** Figure 5 shows $S_{checklist}$ versus total token usage for each method (input and output token breakdowns are in Figure 8 in the Appendix.) End-to-end extraction with GPT-4.1 achieves the highest $S_{checklist}$ of 46.9 but uses 4.4M tokens. GAVEL-AGENT with 26 individual agents using Qwen3 30B-A3B achieves the second-best $S_{checklist}$ of 43.5 while using only 2.8M tokens. This is 36% fewer tokens than end-to-end with GPT-4.1 and 59% fewer than the chunk-by-chunk method with the same Qwen3 model. Within the GAVEL-AGENT configurations, we see a clear quality-cost trade off. A single agent extracting all 26 items is the most token-efficient but provides the lowest $S_{checklist}$. For Qwen3 30B-A3B, the 26-agent configuration achieves the best performance, and the grouped configuration lies in between on both quality and token usage. This shows that, in our setting, agents work better when they focus on fewer items at a time; in the future, being able to reliably handle multiple items per read could unlock further token savings. The best chunk-by-chunk performance is 38.8 with Qwen3 30B-A3B, which is much lower than end-to-end and GAVEL-AGENT. Overall, these results show strong potential for autonomous agents to process long-context inputs, delivering substantially better efficiency while achieving competitive top-level performance. Notably, all document extraction methods fall well below the 68.2 achieved by GPT-5 extracting from human summaries in GAVEL-REF, showing significant headroom for improving both long-context models and long-horizon agents.

## 5 RELATED WORK

**Legal Summarization.** Several datasets exist for this task. Shukla et al. (2022) release Indian and UK Supreme Court cases with human-written summaries, and Elaraby & Litman (2022) provide Canadian court opinions paired with expert summaries. Heddaya et al. (2024) collect U.S. Supreme

Court opinions with their official summaries. These resources focus on single-document summarization with inputs under 16K tokens. Multi-LexSum (Shen et al., 2022) and ExpertLongBench (Ruan et al., 2025) extend this to multi-document summaries using cases from the Civil Rights Litigation Clearinghouse (CRLC), a widely used platform that offers free access to U.S. civil rights cases. Following them, we also collect cases from CRLC, focusing on 2025 filings to reduce data contamination. To better evaluate long-context capability, we construct five length ranges (32K–512K tokens) and benchmark 12 state-of-the-art LLMs with our framework GAVEL-REF, which provides fine-grained analysis of their strengths and weaknesses in long-context legal summarization.

**Checklist-based Evaluation.** With modern LLMs, text evaluation has moved from n-gram metrics such as BLEU (Papineni et al., 2002) or ROUGE (Lin, 2004) to LLM-based methods. One line of work (Min et al., 2023; Scirè et al., 2024) extracts atomic facts from the summaries, and verifies each fact's correctness. While precise, it is limited by inconsistent definitions of what constitutes an 'atomic' fact (Hu et al., 2024) and by poor scalability to long texts. Another line (Lee et al., 2024; Qin et al., 2024; Lin et al., 2025; Cook et al., 2024; Furuhashi et al., 2025) uses LLMs to generate task-specific rubrics and then evaluates responses against each rubric item. In domain-specific settings, human experts often design checklists that capture key information; for example, Arora et al. (2025) ask physicians to write rubrics for medical conversations. The most relevant work, ExpertLongBench (Ruan et al., 2025), introduces expert-designed checklists for 11 tasks, including 26 items for legal summarization (e.g., filing dates, court rulings). Building on this, we improve checklist extraction by requiring evidence for each item and introducing list-wise comparison. We further augment checklist evaluation with residual-fact and writing-style assessments to provide a complete picture of summary quality. Finally, we extend checklist extraction directly to case documents, reducing reliance on human summaries when evaluating future superhuman models.

**LLM Agent Scaffolds.** Modern LLM agents are designed as autonomous problem-solvers that plan actions and invoke tools in a multi-step loop for tasks such as web browsing (Gur et al., 2023), coding (Yang et al., 2024), or general-purpose reasoning. Several open-source scaffolds have been introduced (Xie et al., 2023; Wang et al., 2025; Lu et al., 2025; Qiu et al., 2025). For long-context processing, recent approaches segment documents into chunks or convert them into graph structures (Chen et al., 2023; Sun et al., 2024; Li et al., 2024; Zhao et al., 2024; Zhang et al., 2024), which we adopt as our chunk-by-chunk method. Inspired by how human experts read legal case documents—skimming titles, prioritizing files, and searching for keywords rather than reading everything exhaustively—we develop GAVEL-AGENT, an autonomous scaffold that equips models with six tools for navigating case documents. For context management, unlike the standard approach of continually appending tool calls and responses, we update a snapshot after each tool call and prompt the LLM with it. This design helps maintain an up-to-date state within context limits, especially when models issue 50+ tool calls in sequence, which would otherwise exhaust context quickly.

## 6    CONCLUSION

We present GAVEL-REF, a reference-based framework for evaluating long-context legal summarization that improves checklist-based evaluation with multi-value and support text extraction, and adds residual fact assessment and writing-style evaluation. In our systematic study of 12 frontier LLMs with GAVEL-REF on 2025 cases ranging from 32K to 512K tokens, we find that even the top models—Gemini 2.5 Pro, Claude Sonnet 4, Gemini 2.5 Flash—reach only about 50 $S_{\text{GAVEL-REF}}$, highlighting the difficulty of legal summarization. Our analysis reveals consistent patterns: models perform well on simple single-value items but struggle with multi-value and rare ones, showing key areas for improvement. To reduce reliance on human summaries, we also explore checklist extraction directly from case documents. Comparing end-to-end, chunk-by-chunk, and our proposed GAVEL-AGENT approach, we find that end-to-end extraction with GPT-4.1 achieves the best performance, while GAVEL-AGENT with Qwen3 comes very close and reduces token usage by 36–59%. Looking ahead, advancing long-context models and long-horizon agents for legal summarization and document-level extraction is key to making AI more effective in legal practice.

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

## A    Large Language Model Usage in Paper Writing

We use LLMs solely for language polishing purposes: grammar correction and paraphrasing to improve clarity and readability. We do not use LLMs to generate new content. All semantic content and scientific contributions originate entirely from the authors.

## B    Checklist Definitions

The following are the definitions of the 26 checklist items used in our work, which are adapted from ExpertLongBench (Ruan et al., 2025). We group them into 9 groups.

**A.  Basic Case Information**

1. **Filing Date**: The date when the lawsuit was first initiated with the court.
2. **Parties**: Description of each plaintiff and defendant involved, including relevant positions or offices held. Use specific terms (e.g., "The city", "The parents") rather than generic terms (e.g., "The defendant", "The plaintiffs").
3. **Class Action or Individual Plaintiffs**: Whether the case involves class action plaintiffs or individual plaintiffs with descriptions.
4. **Type of Counsel**: The type(s) of counsel representing each side. Use brief category labels (e.g., private counsel, public interest nonprofit, government counsel, pro se) and include specific organizations (if applicable) in parentheses (e.g., Public interest nonprofit (ACLU)).

**B.  Legal Foundation**

5. **Cause of Action**: The legal vehicle(s) used to bring the claims (the "how" of suing), such as statutes that create a private/enforcement right of action (e.g., 42 U.S.C. § 1983, Title II ADA, FTCA) or judge-made vehicles (e.g., Bivens).
6. **Statutory/Constitutional Basis**: The substantive rights and sources of law allegedly violated (the 'what' was violated), such as specific constitutional provisions/clauses (e.g., Fourteenth Amendment—Equal Protection, First Amendment—Freedom of Association, Eighth Amendment) and statutory rights (e.g., ADA Title II, Rehab Act § 504).
7. **Remedy Sought**: What each party asks the court to grant, not what the court ordered or what the parties settled. Include both sides if the defendant seeks relief.

**C.  Judge Information**

8. **Judge Name**: The first and last name of the judge(s) involved in the case. Do not include Supreme Court Justices.

**D.  Related Cases**

9. **Consolidated Cases**: Cases that were combined with this case for joint proceedings.
10. **Related Cases**: Other cases referenced or connected to this case, listed by case code number.

**E.  Filings and Proceedings**

11. **Important Filings**: Significant motions filed, including temporary restraining orders, preliminary injunctions, motions to dismiss, and motions for summary judgment.
12. **Court Rulings**: Judicial decisions on important filings such as motions to dismiss, summary judgment, preliminary injunctions, class certification, and attorneys' fees (excluding amended complaints and statements of interest).
13. **Reported Opinions**: Citations of reported opinions using shortened Bluebook format (e.g., "2020 WL 4218003"), without case name, court, or date unless from a different case.
14. **Trials**: Information about trial proceedings including scheduling, outcomes, and related motions or rulings.
15. **Appeals**: Whether appeals were filed, which parties appealed, to which court, and the outcomes.

**F.  Decrees**

16. **Significant Terms**: The substantive obligations ordered by the court. This includes consent decrees and stipulated judgments/injunctions because they are entered as court orders.

17. **Decree Dates**: All decree-related dates such as entry date, modification/amendment dates (of the order), suspension/stay dates, partial termination dates, and full termination/vacatur dates. Decrees include injunctions, consent decrees, or stipulated judgments/injunctions.
18. **Duration**: The duration of all decrees obligations (each as a separate entry). A 'decree' is any formal order or judgment issued by a court such as an injunction, consent decree, or stipulated judgment/injunction, as opposed to a negotiated agreement between parties.

**G. Settlements**

19. **Settlement Terms**: The substantive obligations the parties agree to in a settlement that is not entered as a court order. A settlement may be court-approved or enforced, but as long as it is not entered as an order, it is a settlement.
20. **Settlement Date**: All settlement-related dates (each as a separate entry) such as execution/signing date(s), court approval date (if approved but not entered as an order), amendment dates, enforcement/retention dates without incorporation (e.g., court retains jurisdiction over the settlement but does not enter it as an order), and termination/expiration of the settlement agreement (if contractual).
21. **Duration**: The duration of all settlements obligations (each as a separate entry). A 'settlement' is any negotiated agreement between parties that resolves a dispute, as opposed to a formal order or judgment issued by a court.
22. **Court Enforcement**: Whether the settlement (not entered as an order/judgment) is court-enforced. Answer Yes if the court explicitly retains jurisdiction to enforce the settlement without incorporating it into an order/judgment (e.g., Kokkonen retention). Answer No if it's a private agreement with no retained jurisdiction.
23. **Enforcement Disputes**: The disputes about enforcing a settlement (a negotiated agreement not entered as a court order)—e.g., motions to enforce/contempt or requests invoking retained jurisdiction—each as a separate value with date, movant, issue, and outcome (or pending).

**H. Monitoring**

24. **Monitor Name**: Name of any court-appointed monitor or special master.
25. **Monitor Reports**: Monitor's findings regarding defendant compliance with court orders, including which terms are being met.

**I. Context**

26. **Factual Basis**: The underlying facts and evidence supporting the legal claims, including: (i) details of relevant events (what, when, where, who), (ii) supporting evidence (physical, documentary, testimonial), and (iii) background context.

## C  WRITING STYLE SIMILARITY EVALUATION DETAILS

The following are the definitions of the five aspects used in our writing style similarity evaluation. Each aspect is rated on a 1–5 Likert scale, where 5 indicates identical and 1 indicates completely different.

1. **Readability & Jargon Level**

   Compare the reading level and the balance of legal jargon vs. plain language. Consider terminology density and accessibility to non-legal readers.

   5 Nearly identical reading level and jargon density; same balance of technical/plain language throughout.
   4 Very similar complexity with minor differences in terminology or occasional variance in technical language.
   3 Moderate differences in accessibility; one is noticeably more technical in places but overall similar.
   2 Significantly different complexity; one is consistently more technical or more accessible.
   1 Completely different target audiences (e.g., one for legal professionals, the other for the general public).

2. **Narrative Order**

   Compare whether events are presented in the same sequence (chronological vs. thematic) and the ordering of key facts and arguments.

    5 Identical sequence of information; same events, facts, and arguments in the same order.

    4 Same overall flow with 1–2 elements reordered; core structure preserved.

    3 Similar general structure but several sections reordered; recognizable yet rearranged.

    2 Different organizational approaches with some overlap (mix of chronological and thematic).

    1 Completely different information architecture (e.g., one chronological, the other organized by issues).

3. **Sentence Structure & Voice**
   Compare sentence variety, active vs. passive voice, and tense consistency.

    5 Nearly identical sentence patterns, voice usage, and tense choices throughout.

    4 Very similar style with occasional differences in sentence complexity or voice.

    3 Moderate variation; one favors longer/shorter sentences or more active/passive constructions.

    2 Noticeably different styles; consistent differences in sentence variety and voice preferences.

    1 Completely different approaches (e.g., one varied and active; the other uniform and passive).

4. **Formatting & Layout**
   Compare use of headings, bullet/numbered lists, paragraphing, and other structural cues.

    5 Identical formatting choices; same use of headings, lists, and paragraph breaks.

    4 Very similar structure with minor variations (e.g., one extra heading or different list style).

    3 Similar approach but noticeable differences in execution (e.g., both use headings but at different levels/frequency).

    2 Different formatting philosophies; one is much more structured than the other.

    1 Completely different (e.g., one heavily formatted; the other continuous prose).

5. **Citation & Reference Style**
   Compare presence, position, and formatting of case/statute citations or footnotes (inline vs. separate), citation density, and conventions.

    5 Identical citation approach; same style, frequency, and positioning.

    4 Very similar practices with minor formatting differences or occasional variation in placement.

    3 Similar philosophy but different execution (e.g., both cite cases but differ in density/positioning).

    2 Different approaches; one is substantially more reference-heavy or uses a different citation style.

    1 Completely different or incomparable (e.g., one with extensive citations, the other with none).

## D  ANNOTATION DETAILS

**Annotator Recruitment.**  We recruit four in-house annotators who are native English speakers and U.S.-based undergraduate students with basic familiarity with legal cases. All annotators are trained by the authors: we review the 26 checklist items together, ensure that everyone understands the legal terms involved (e.g., decree, settlement, ruling), and walk through example annotations. Because their task is to extract checklist items from case summaries that are written for lay readers rather than to provide legal judgments or read case documents, we do not require formal legal training once they clearly understand each checklist item and its definition.

**Inter-Annotator Agreement.**  For checklist extraction, the ten longest summaries receive triple annotations. Agreement is measured as the average pairwise $S_{\text{checklist}}$ score across annotators, reaching 83.6 (using Gemma3 27B as the comparison model). For checklist comparison, single-value pairs achieve moderate agreement with Fleiss' $\kappa = 0.57$, while multi-value matching yields an average pairwise F1 of 0.82, indicating high consistency. For writing style similarity, Krippendorff's $\alpha$ (Krippendorff, 2011) across the five aspects averages 0.32. We also measure a "two-agree" metric: overall, at least two annotators agree with each other on the rating 94.4% of the time, and all three annotators choose different ratings only 5.6% of the time. This indicates that most instances of writing-style rating show clear majority agreement, and full disagreement is rare.

**Annotation Interfaces.**  Figures 23, 24, and 25 display screenshots of our human annotation interfaces for checklist extraction, checklist comparison, and writing style similarity rating, respectively.

The collected data are used for the meta-evaluation of GAVEL-REF and for evaluating checklist extraction from case documents methods.

## E  FURTHER ANALYSIS

Figure 6 shows the average summary length of each LLM in each case-length bin, alongside the overall $S_{\text{GAVEL-REF}}$ score.

Compared to human summaries, LLMs only approach human length in the 32K–128K bins; for 256K and 512K cases, all models produce much shorter summaries than humans. In general, open-source models generate noticeably shorter summaries than proprietary models. Among all models, GPT5 is an outlier: it consistently produces very long summaries (often over 900 words) even for short cases (32K–128K), substantially longer than the human references. Figure 9 shows a typical example. GPT-5 often writes in a highly verbose, list-style format rather than a narrative, which we hypothesize is related to its "high" thinking mode. We also compute instance-level correlations between summary length and $S_{\text{GAVEL-REF}}$. Overall, we observe a moderate positive correlation (Pearson $r = 0.31$, Spearman $\rho = 0.36$, Kendall's $\tau = 0.24$), but this is largely driven by weaker open-source models that both underperform and produce shorter summaries. When

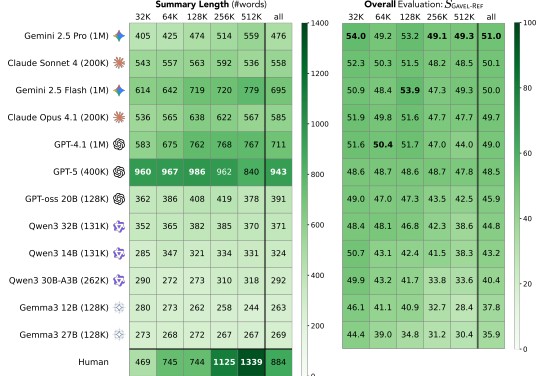

Figure 6: Summary length and overall evaluation for 12 LLMs. As case length increases, all models perform worse. For the cases in the 256K and 512K bins, LLM-generated summaries are much shorter than human summaries and fail to include as much information.

we separate proprietary and open-source models, the correlations become much smaller: within proprietary models, Pearson $r = -0.11$, Spearman $\rho = -0.13$, and Kendall's $\tau = 0.09$; within open-source models, Pearson $r = 0.20$, Spearman $\rho = 0.20$, and Kendall's $\tau = 0.14$. This suggests that, once we control for model family, summary length alone explains only a small fraction of the performance differences.

Figure 7 presents the item-level performance for the top 3 models in checklist evaluation—Gemini 2.5 Flash, Pro and Claude Sonnet 4—showing their top and bottom 5 checklist items plus consistently over- and under-specified items. All three models exhibit high similar performance patterns across items.

Figure 8 presents the checklist extraction performance $S_{\text{checklist}}$ versus total, input, output token usage for each method extracting checklist from case documents.

Figures 10, 11, and 12 present the checklist item-level performance for each of the 12 LLMs we evaluate.

Figures 13 to 22 show a randomly sampled case, comparing checklists extracted directly from case documents by GAVEL-AGENT with Qwen3 30B-A3B (26-agent configuration) against the human-annotated checklist extracted from the case summary.

## F  IMPLEMENTATION DETAILS

For all language models, we use a temperature of 0.7 and top-p of 1, except for GPT-5 (where temperature cannot be changed and is fixed at 1) and Qwen3, for which we use a temperature of 0.6 and top-p of 0.95, following the official recommendations. For Gemini 2.5 Flash and Pro, we set the thinking budget to -1 (allowing the model to decide). For GPT-5, we use "high" thinking effort. For Claude Sonnet 4 and Opus 4.1, we set the thinking budget to 10,000.

We use the following versions of the proprietary models: gpt-4.1-2025-04-14, gpt-5-2025-08-07, claude-sonnet-4-20250514, claude-opus-4-1-20250805, gemini-2.5-flash (June 2025), and gemini-

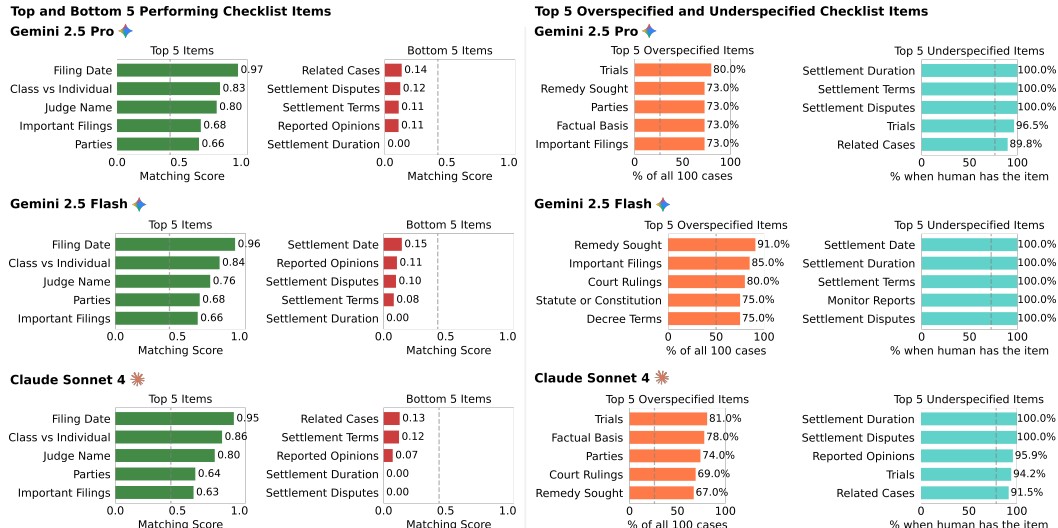

Figure 7: Performance breakdown for the top-3 models in checklist evaluation (Gemini 2.5 Pro, Gemini 2.5 Flash, and Claude Sonnet 4): top/bottom 5 checklist items by matching score and most frequently over/under-specified items. Overspecification measured as frequency across all 100 cases; underspecification as frequency among cases where human summary includes that item.

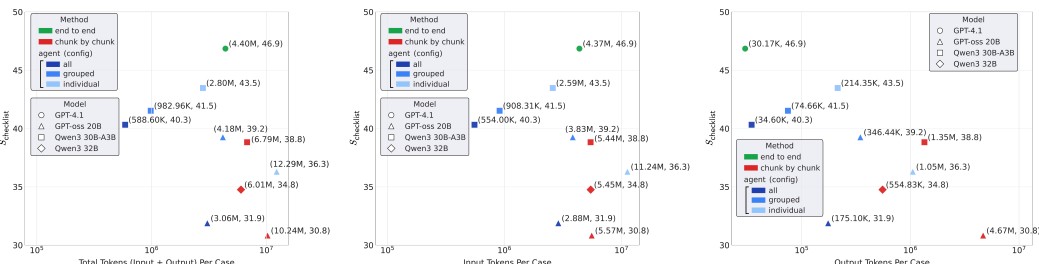

Figure 8: $S_{\text{checklist}}$ versus total token, input token, and output token usage for different methods extracting from case documents.

2.5-pro (June 2025). For open-source models, we use the instruction-tuned version of Gemma3 (Gemma3-it) and Qwen3-30B-A3B-Thinking-2507 for Qwen3 30B-A3B. Open-source models are run through vLLM Kwon et al. (2023) on 4 A40 GPUs. For all reasoning models such as Qwen3, we use the reasoning mode. Due to compute constraints, we could not run models larger than these, such as GPT-oss 120B. The total API costs is $1,800 USD.

For GAVEL-AGENT, we implement tool calls using each model's native format: ChatML for Qwen3 and Harmony for GPT-oss.

## G PROMPTS

The following lists the prompts used in our paper.

**Prompts used in GAVEL-REF.** Figure 26 shows the prompt for extracting checklist items from summaries. Figures 27 and 28 show the prompts for comparing single-value and multi-value checklist items, respectively. Figure 29 shows the prompt for extracting residual facts not covered by checklist items or their supporting text. Figure 30 shows the prompt for rating writing style similarity between two summaries across five aspects.

**Prompt for summarization.** Figure 31 shows the prompt for legal summarization.

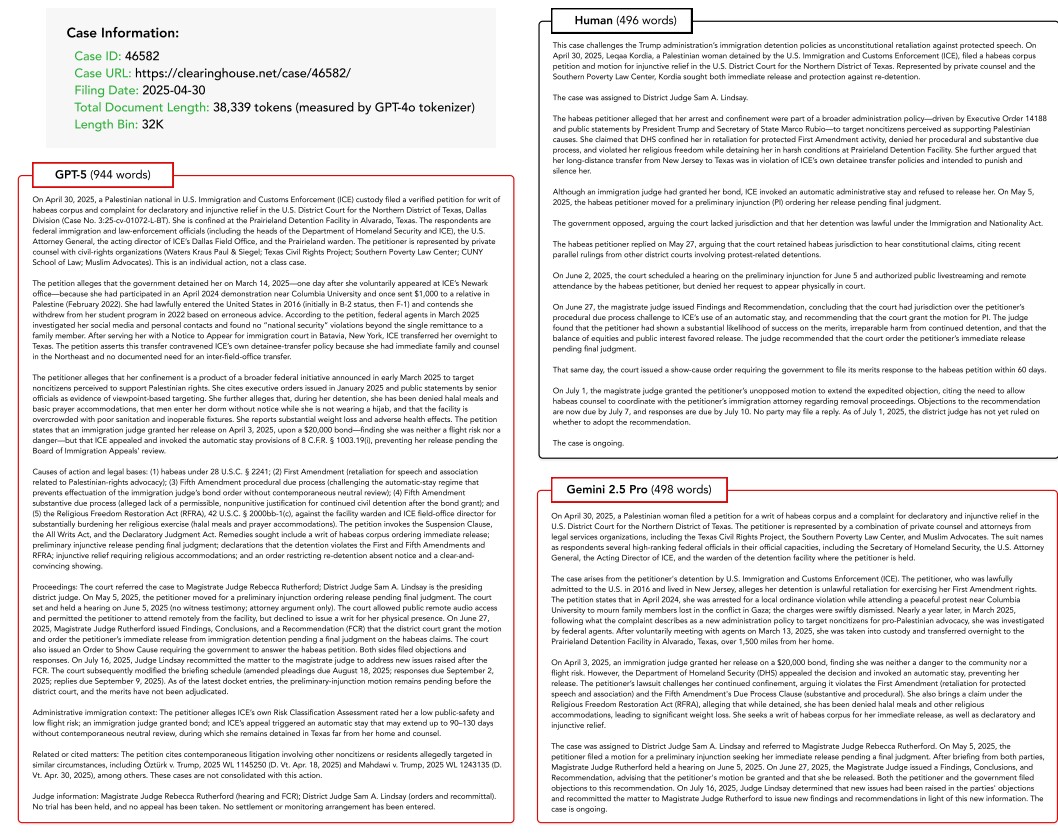

Figure 9: Example summaries from GPT-5, Gemini 2.5 Pro, and a human reference for a case in the 32K bin. This illustrates why GPT-5 produces very long summaries (as seen in Figure 6) even for short cases.

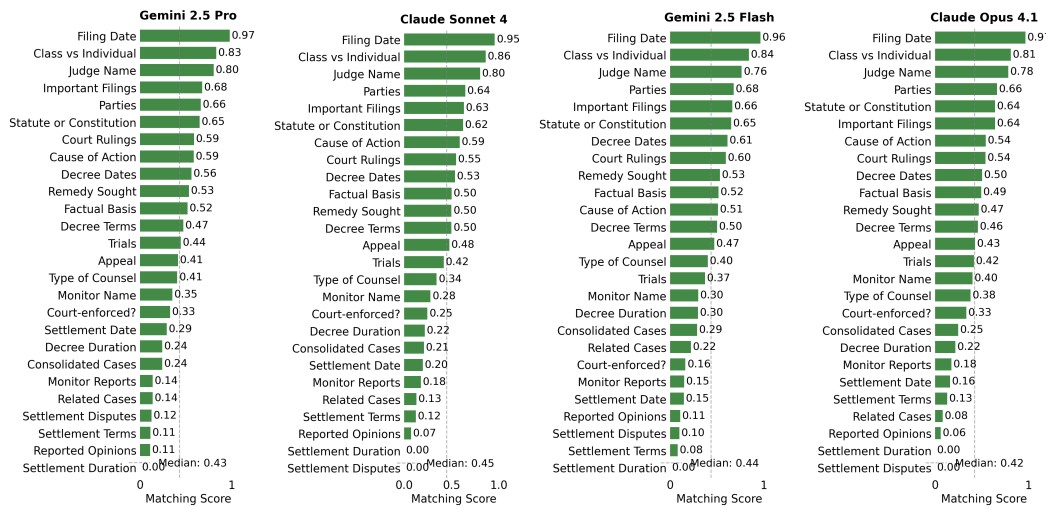

Figure 10: Checklist item-level performance for each LLM in the checklist evaluation. The metric is the matching score $m_i$. This figure shows results for Gemini 2.5 Pro, Claude Sonnet 4, Gemini 2.5 Flash, and Claude Opus 4.1.

**Prompts for checklist extraction from case documents.** Figures 32 and 33 present the prompts for the end-to-end method. Figure 34 presents the prompt for the chunk-by-chunk method. Figures 35, 36, and 37 present the system prompts used in GAVEL-AGENT.

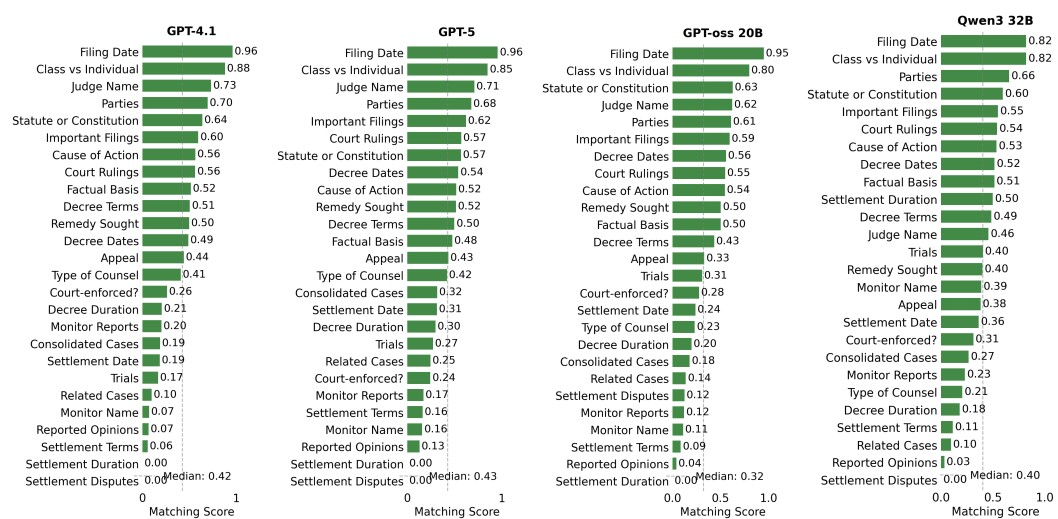

Figure 11: Checklist item-level performance for each LLM in the checklist evaluation. The metric is the matching score $m_i$. This figure shows results for GPT-4.1, GPT-5, GPT-oss 20B, Qwen3 32B.

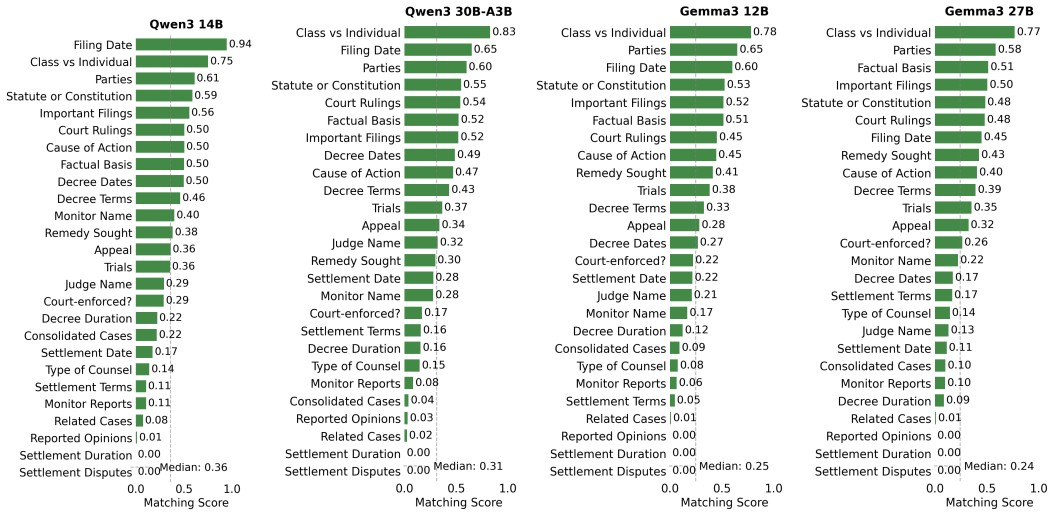

Figure 12: Checklist item-level performance for each LLM in the checklist evaluation. The metric is the matching score $m_i$. This figure shows results for Qwen3 14B, Qwen3 30B-A3B, Gemma3 12B and Gemma3 27B.

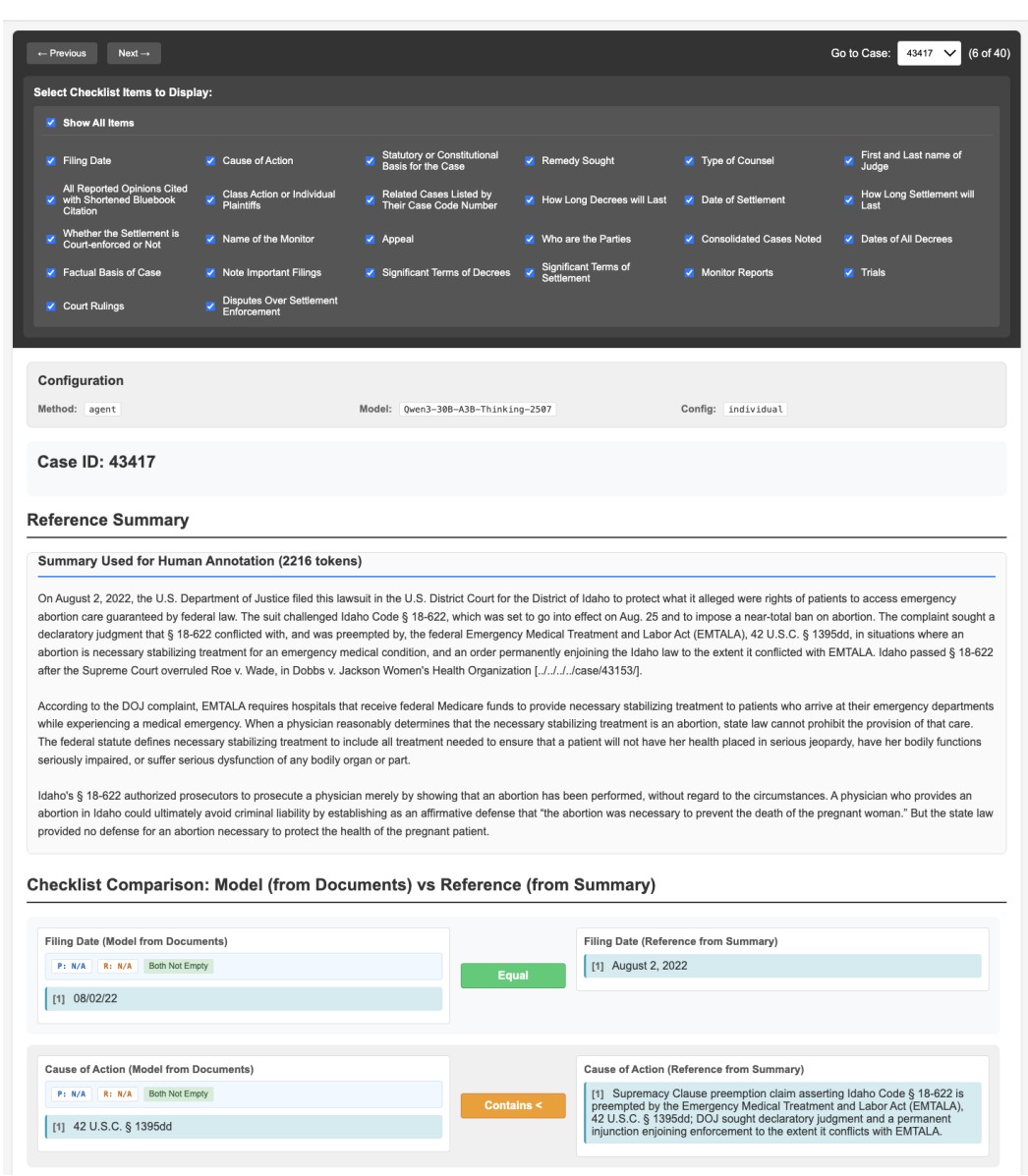

Figure 13: Screenshot of a visualization for one case, comparing checklists extracted directly from case documents by GAVEL-AGENT with Qwen3 30B-A3B (26 individual agents configuration) against the human-annotated checklist extracted from the case summary (figure 1 of 10).

**Statutory or Constitutional Basis for the Case (Model from Documents)**

P: 0.50   R: 1.00   Both Not Empty

[1] Emergency Medical Treatment and Labor Act (EMTALA), 42 U.S.C. § 1395dd

[2] Supremacy Clause (U.S. Const. art. VI, cl. 2)

**Partial Match**

**Statutory or Constitutional Basis for the Case (Reference from Summary)**

[1] U.S. Constitution, Supremacy Clause (preemption)

**Remedy Sought (Model from Documents)**

P: 0.33   R: 0.67   Both Not Empty

[1] Declaratory judgment stating that Idaho Code § 18-622 violates the Supremacy Clause and is preempted and therefore invalid to the extent that it conflicts with EMTALA

[2] Declaratory judgment stating that Idaho may not initiate a prosecution against, seek to impose any form of liability on, or attempt to revoke the professional license of any medical provider based on that provider's performance of an abortion that is authorized under EMTALA

[3] Preliminary and permanent injunction against the State of Idaho—including all of its officers, employees, and agents—prohibiting enforcement of Idaho Code § 18-622(2)-(3) to the extent that it conflicts with EMTALA

[4] Any and all other relief necessary to fully effectuate the injunction against Idaho Code § 18-622's enforcement to the extent it conflicts with EMTALA

[5] The United States' costs in this action

[6] Any other relief that the Court deems just and proper

**Partial Match**

**Remedy Sought (Reference from Summary)**

[1] Declaratory judgment that Idaho Code § 18-622 conflicts with and is preempted by EMTALA when an abortion is necessary stabilizing treatment for an emergency medical condition.

[2] Permanent injunction enjoining enforcement of Idaho Code § 18-622 to the extent it conflicts with EMTALA.

[3] Preliminary injunction prohibiting enforcement of Idaho Code § 18-622 as applied to EMTALA-mandated care.

**Type of Counsel (Model from Documents)**

P: 1.00   R: 1.00   Both Not Empty

[1] Plaintiff: Government counsel (U.S. Department of Justice)

[2] Defendant: Government counsel (State of Idaho)

**Equal**

**Type of Counsel (Reference from Summary)**

[1] Federal government counsel (U.S. Department of Justice)

[2] State government counsel (Idaho Acting Solicitor General)

**First and Last name of Judge (Model from Documents)**

P: N/A   R: N/A   Both Not Empty

[1] B. Lynn Winmill

**Contains <**

**First and Last name of Judge (Reference from Summary)**

[1] Full name: "Judge Barry Lynn Winmill"

Figure 14: Screenshot of a visualization for one case, comparing checklists extracted directly from case documents by GAVEL-AGENT with Qwen3 30B-A3B (26 individual agents configuration) against the human-annotated checklist extracted from the case summary (figure 2 of 10).

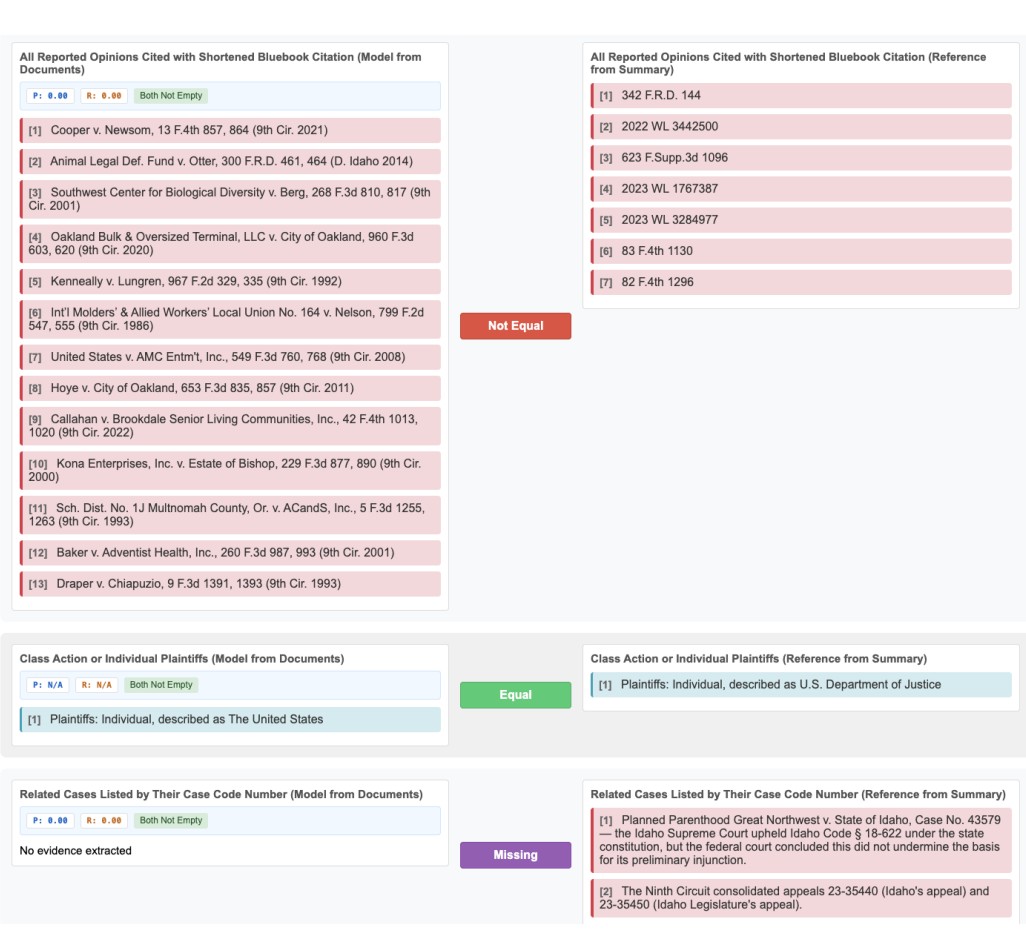

Figure 15: Screenshot of a visualization for one case, comparing checklists extracted directly from case documents by GAVEL-AGENT with Qwen3 30B-A3B (26 individual agents configuration) against the human-annotated checklist extracted from the case summary (figure 3 of 10).

Figure 16: Screenshot of a visualization for one case, comparing checklists extracted directly from case documents by GAVEL-AGENT with Qwen3 30B-A3B (26 individual agents configuration) against the human-annotated checklist extracted from the case summary (figure 4 of 10).

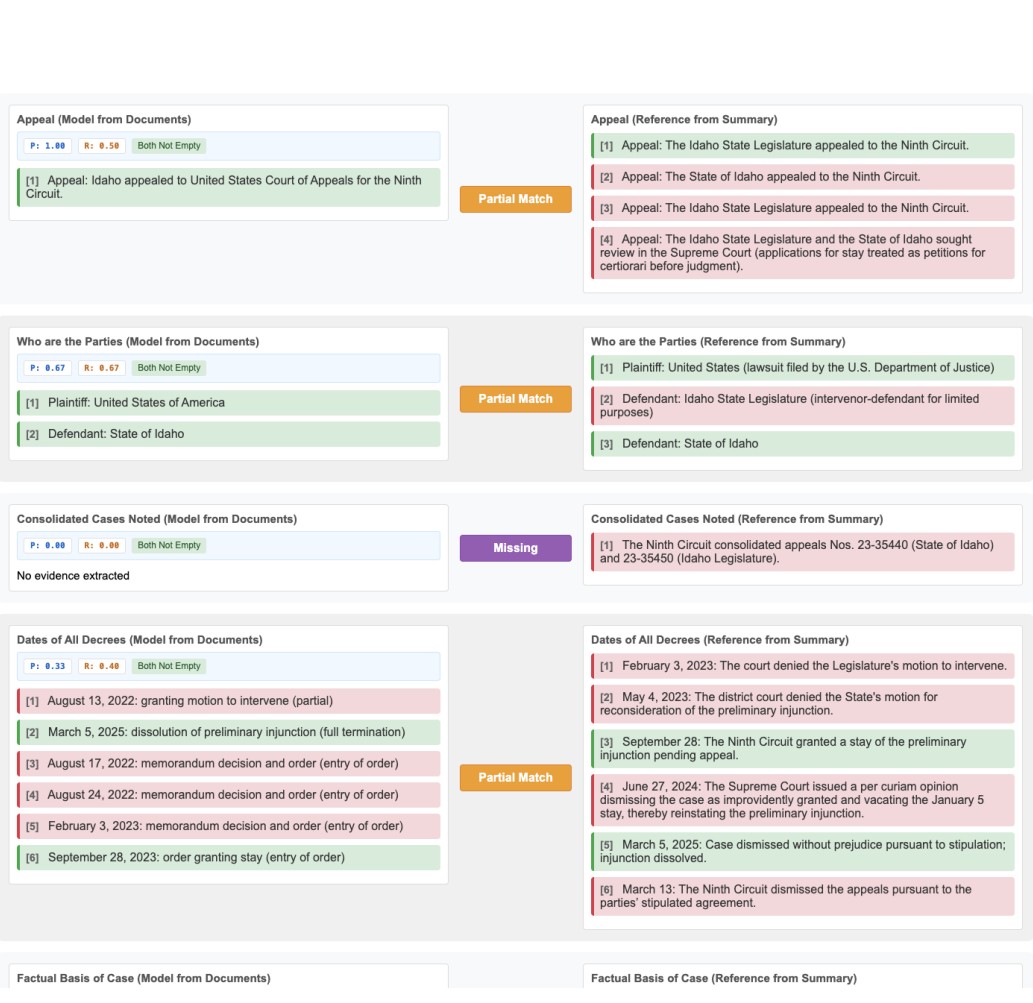

Figure 17: Screenshot of a visualization for one case, comparing checklists extracted directly from case documents by GAVEL-AGENT with Qwen3 30B-A3B (26 individual agents configuration) against the human-annotated checklist extracted from the case summary (figure 5 of 10).

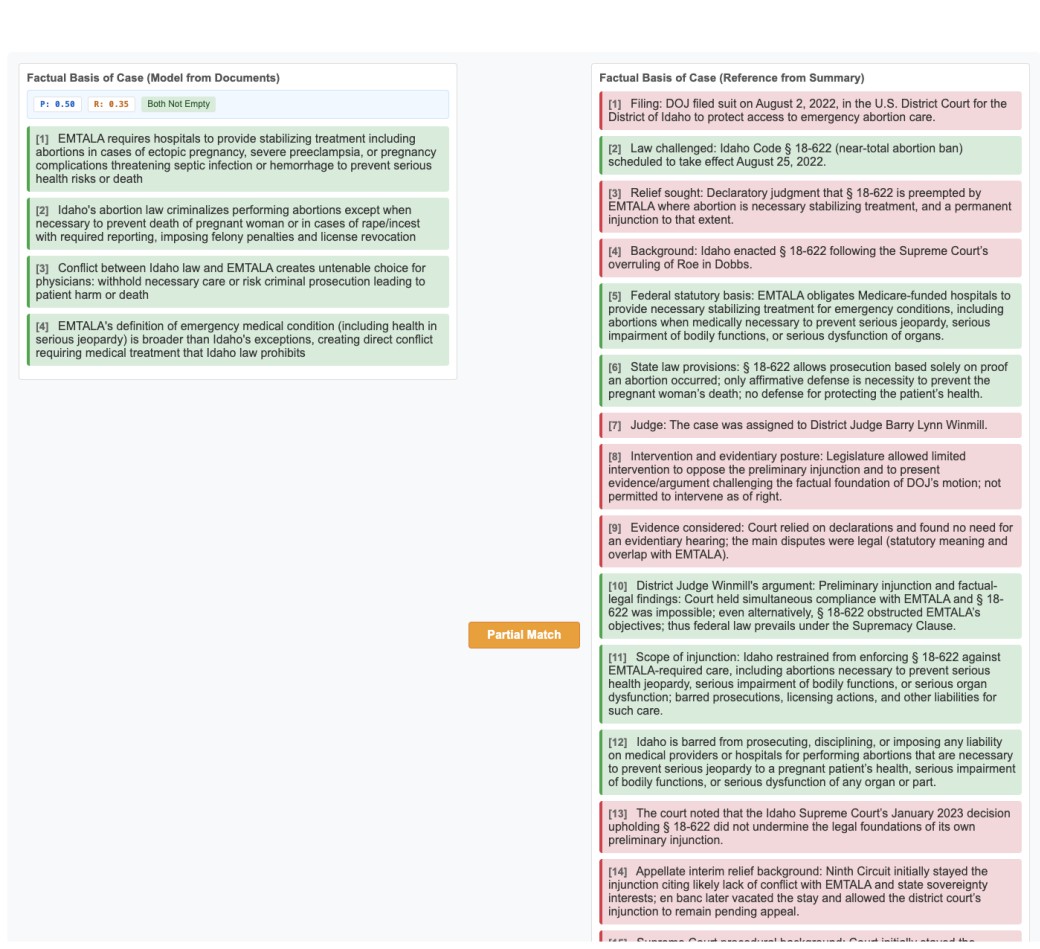

Figure 18: Screenshot of a visualization for one case, comparing checklists extracted directly from case documents by GAVEL-AGENT with Qwen3 30B-A3B (26 individual agents configuration) against the human-annotated checklist extracted from the case summary (figure 6 of 10).

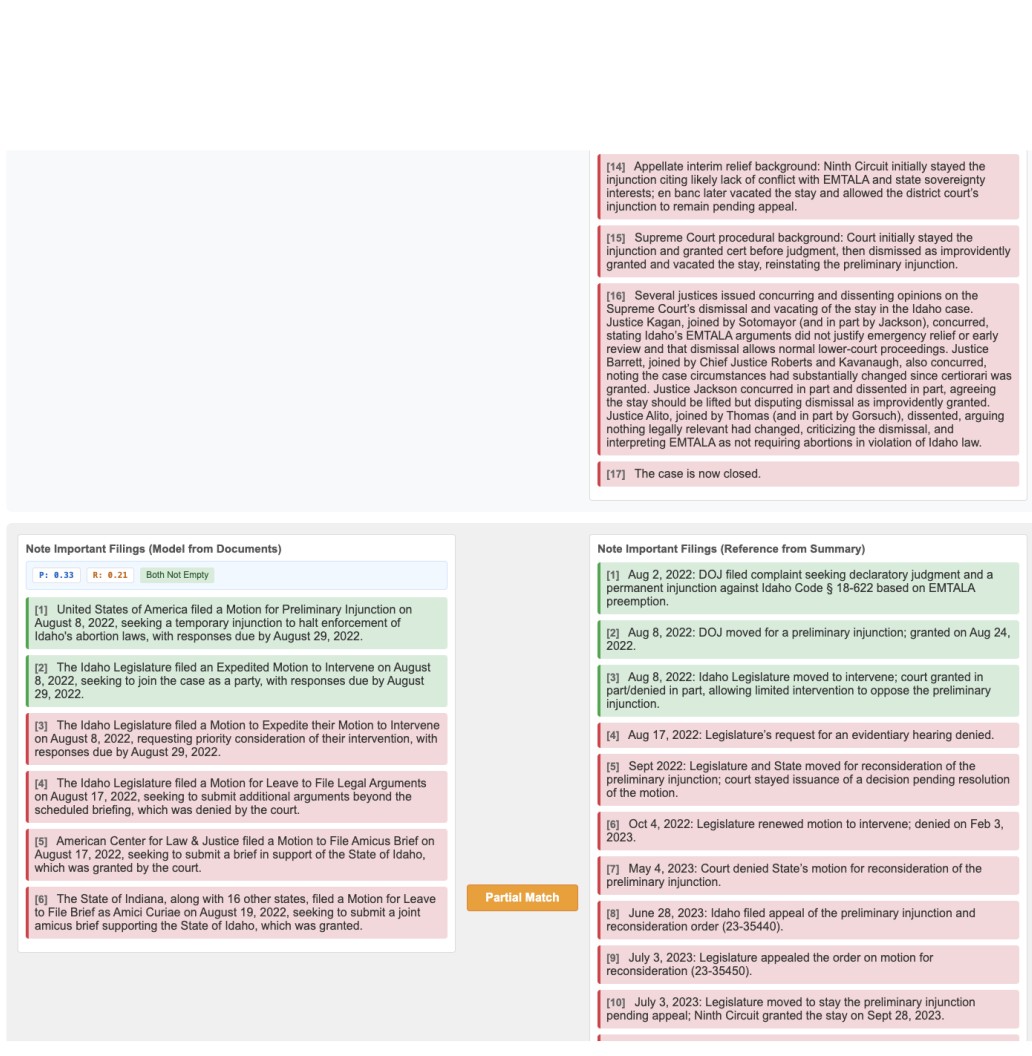

Figure 19: Screenshot of a visualization for one case, comparing checklists extracted directly from case documents by GAVEL-AGENT with Qwen3 30B-A3B (26 individual agents configuration) against the human-annotated checklist extracted from the case summary (figure 7 of 10).

Figure 20: Screenshot of a visualization for one case, comparing checklists extracted directly from case documents by GAVEL-AGENT with Qwen3 30B-A3B (26 individual agents configuration) against the human-annotated checklist extracted from the case summary (figure 8 of 10).

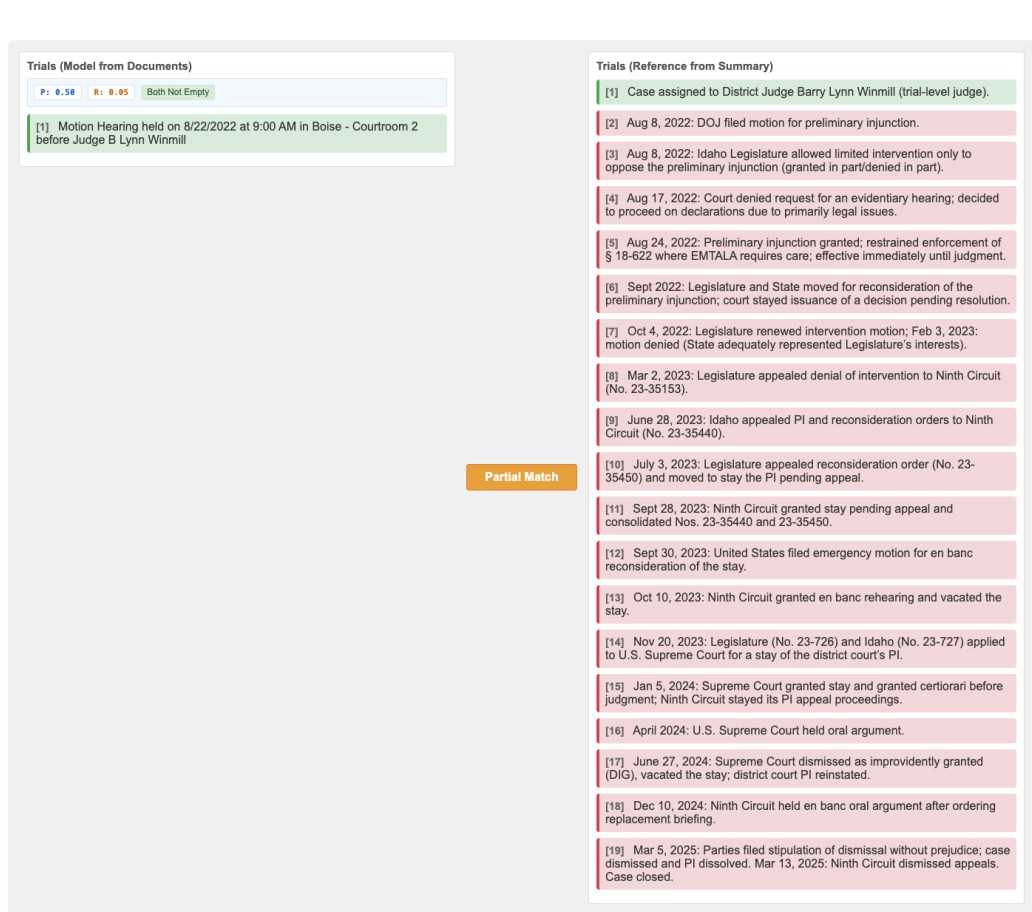

Figure 21: Screenshot of a visualization for one case, comparing checklists extracted directly from case documents by GAVEL-AGENT with Qwen3 30B-A3B (26 individual agents configuration) against the human-annotated checklist extracted from the case summary (figure 9 of 10).

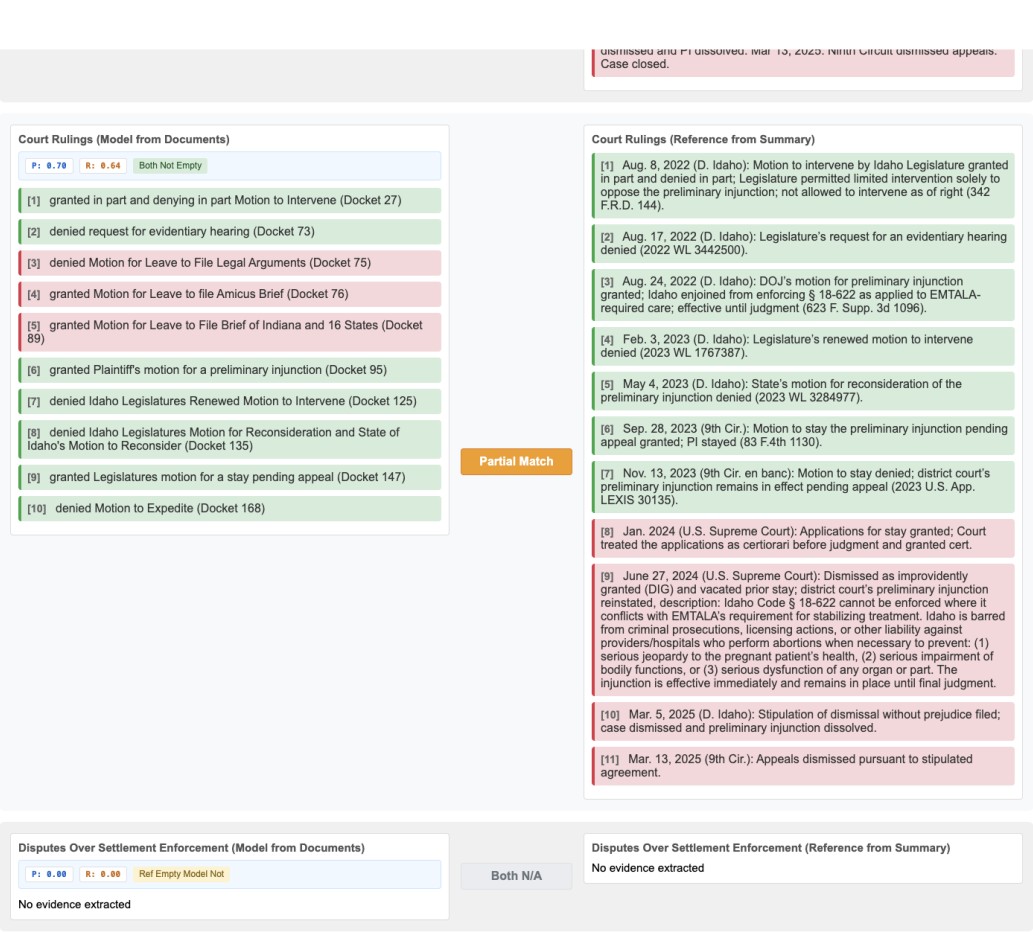

Figure 22: Screenshot of a visualization for one case, comparing checklists extracted directly from case documents by GAVEL-AGENT with Qwen3 30B-A3B (26 individual agents configuration) against the human-annotated checklist extracted from the case summary (figure 10 of 10).

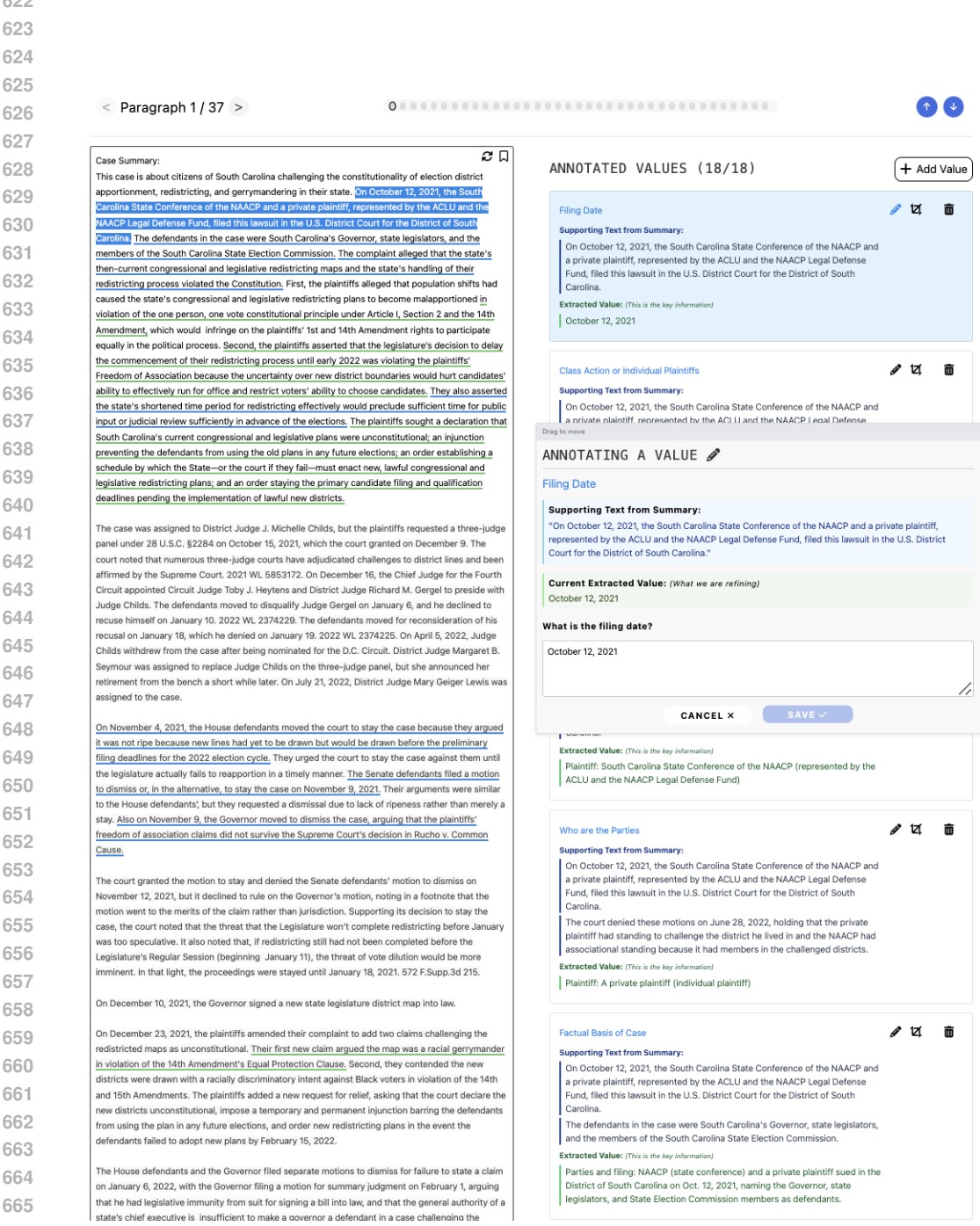

Figure 23: Screenshot of the annotation interface for checklist extraction from summaries. Annotators can add, remove, or modify checklist item values, with the process carried out paragraph by paragraph to ensure each sentence is carefully reviewed.

## Legal Case Summary Checklist Comparison

Instance 1 of 110
Time: 4:30

Welcome, user1

Logout

**Task Instructions**

You are comparing two lists of legal information about **Dates of All Decrees**. Your task is to match semantically equivalent items between the two lists by dragging items from List B to match with items in List A.

- Click and drag items from List B to the matching item in List A
- Items may be paraphrased or formatted differently but convey the same meaning
- Some items may not have matches - that's okay
- Click on a matched pair to unmatch them

**Case ID: 46341 | Category: Dates of All Decrees**

| List A | List B |
|---|---|
| 1. June 23, 2025: Judge Young entered a partial final judgment under Federal Rule of Civil Procedure 54(b), ruling the agency directives and resulting grant terminations arbitrary and capricious under the APA, and vacating and setting aside both the directives and the specific grant terminations affecting the plaintiff states. | 1. May 12: the court issued an order affirming its subject matter jurisdiction. |
| 2. June 24, 2025: The district court denied the government's motion to stay the judgment. | 2. June 16, 2025: the court held a Phase 1 bench trial and ruled in favor of the plaintiffs by vacating the challenged government directives. |
| 3. July 2, 2025: The district court issued a full written opinion (Am. Pub. Health Ass'n v. NIH, 2025 U.S. Dist. LEXIS 125988). | 3. June 23, 2025: the court adopted the plaintiffs' revised proposed judgment, holding the directives and resulting terminations arbitrary and capricious, void, unlawful, and without legal effect; and ordered judgment for plaintiffs on Count Three. |
| 4. July 18, 2025: The First Circuit denied a stay in an opinion (National Institutes of Health v. American Public Health Association, 145 F. 4th 39). | 4. July 18, 2025: the First Circuit denied to stay the district court's judgment pending appeal. |
| 5. August 21, 2025: The U.S. Supreme Court issued a partial stay, staying the portion of the district court's judgment that vacated the individual grant terminations, but denying a stay as to the vacatur of the underlying agency directives (National Institutes of Health v. American Public Health Assn., 606 U.S. ___). | 5. August 21, 2025: the Supreme Court partially granted and partially denied the stay application—staying the district court's judgments vacating the termination of research grants, but denying a stay as to the judgments vacating the NIH guidance documents. |

**Current Matches:**
No matches yet. Drag items from List B to List A to create matches.

**Feedback (Optional)**

Any comments or issues with this instance?

☐ This instance has a problem (e.g., unclear information, formatting issues)

Skip    Submit

Figure 24: Screenshot of the annotation interface for checklist comparison. Annotators match items between two lists in a list-wise comparison. For string-wise comparison, where both values are strings, the middle component becomes a radio selection with four options: equal, A contains B, B contains A, or different.

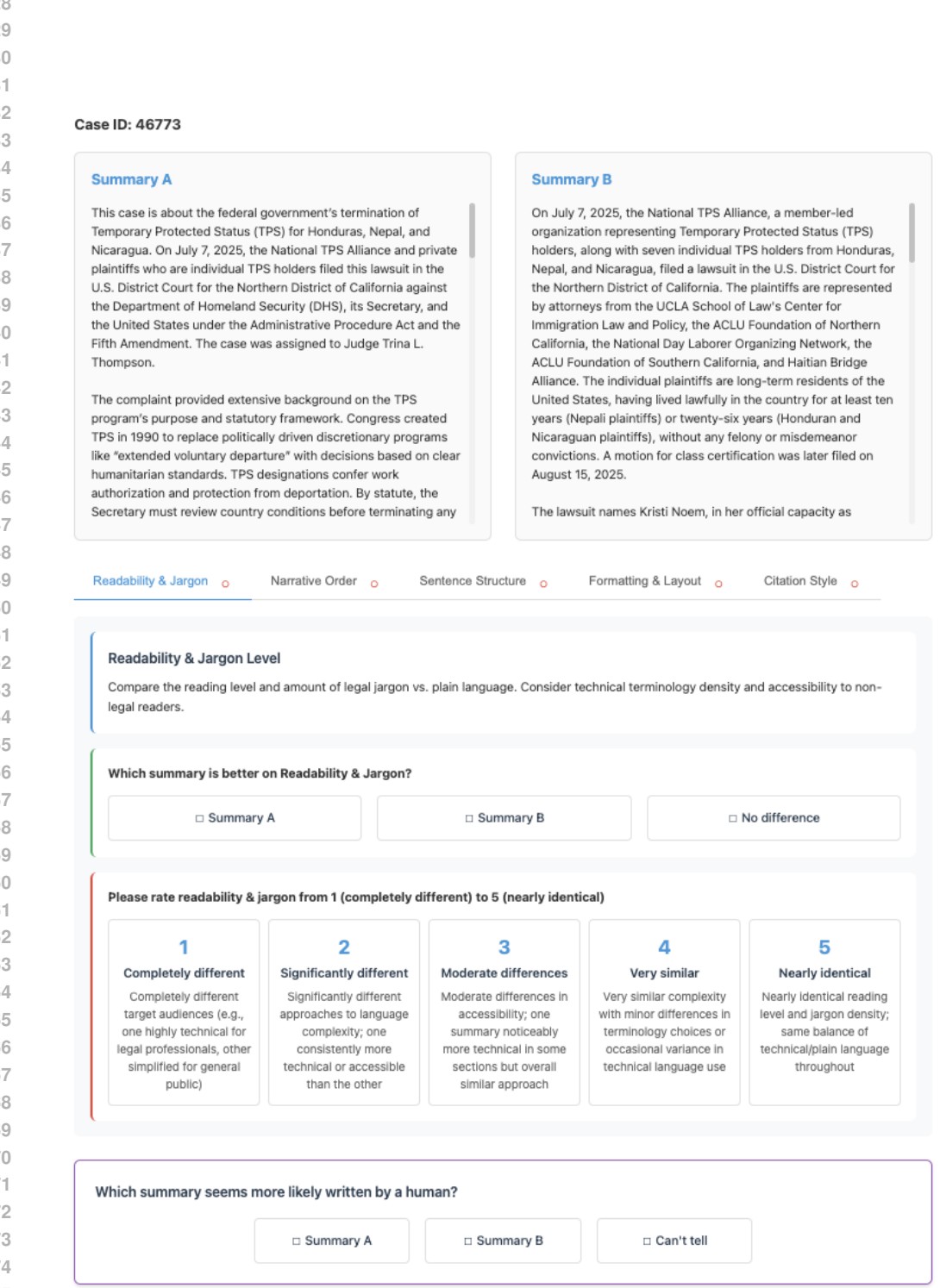

Figure 25: Screenshot of the annotation interface for rating writing style similarity. Annotators compare two summaries, providing ratings on five aspects and answering auxiliary questions such as which summary they prefer.

```
Prompt for Extracting Checklist from Summary

You are assisting a lawyer in extracting key information from a
↪  legal case summary. Given a case summary, identify
↪  {checklist_item_definition}
# Note: Do not make assumptions or add information that is not
↪  presented in the summary.

# Case Summary
{case_summary}

# Output Format
Your output should be in the following JSON format-no extra keys,
↪  no prose outside of the JSON:

```
{{
  "reasoning": "<brief analysis of the case summary and how you
   ↪  identified the relevant information or determined that none
   ↪  was present>",
  "extracted": [
    {{
      "evidence": [
        "<verbatim snippet 1>",
        "<verbatim snippet 2 (if multiple snippets are relevant)>"
        // ...
      ],
      "value": "<extracted information from the evidence>"
    }}
    // ...
  ]
}}
```

## Definitions of each part
- `reasoning`: A brief analysis of the case summary and how you
↪  identified the relevant information or determined that none was
↪  present.
- `extracted`: A list of one or more objects, each representing a
↪  distinct piece of information relevant to the checklist item
↪  (e.g., multiple court rulings, decree dates, or cited
↪  opinions). Always use a list, even if there is only one item.
- `evidence`: One or more exact text snippets copied from the case
↪  summary that support the extracted information. Always return
↪  as a list of strings.
- `value`: The extracted information.

## Rules for the JSON schema
1. **extracted** and **evidence** is always a list, even if they
↪  hold a single object.
2. Copy the **evidence** exactly as it appears in the case
↪  summary-no rewriting.
3. If the case summary contains no relevant information, output the
↪  **extracted** as an empty list:

```
{{
  "reasoning": "<brief analysis>",
  "extracted": []
}}
```
```

Figure 26

```
Prompt for Comparing Single-Value Checklist Item

You are given two pieces of legal information (A and B) about
↪ **{checklist_category}**, extracted from two summaries of
↪ the same case. Your task is to compare these pieces of
↪ information based on their **semantic meaning** – that
↪ is, what they actually convey, regardless of how they are
↪ worded or formatted.

# Information to Compare
## Information A:
{information_A}

## Information B:
{information_B}

# Relationship Options
Determine which of these four relationships best describes
↪ how A and B relate to each other:
1. **"A contains B"** – A includes all the information in B,
↪ plus additional information
2. **"B contains A"** – B includes all the information in A,
↪ plus additional information
3. **"A equals B"** – A and B convey the same information
↪ (semantically equivalent)
4. **"A and B are different"** – A and B contain different or
↪ conflicting information

# Output Format
Structure your response as follows:
**Reasoning:** Provide your detailed analysis of how the two
↪ pieces of information relate to each other

**Final Answer:** State one of the four options: "A contains
↪ B", "B contains A", "A equals B", or "A and B are
↪ different"
```

Figure 27

```
Prompt for Comparing Multi-Value Checklist Item

You are given two lists of legal information (A and B) about
↪  **{checklist_category}**, extracted from two summaries of the
↪  same legal case. Your task is to compare these lists based on
↪  their **semantic meaning**-that is, what each item conveys,
↪  regardless of wording, format, or phrasing.

You should identify:
1. Items that appear in **both A and B** (i.e., semantically
↪  equivalent),
2. Items that appear **only in A**,
3. Items that appear **only in B**.

# Information to Compare
## List A:
{information_A}

## List B:
{information_B}

# Output Format
Structure your response as follows:
**Reasoning:**
Provide your detailed analysis of how the two lists relate to each
↪  other. Explain any mappings between items, and how you
↪  determined whether they were equivalent or different.

**Final Answer:**
Output a valid JSON object with the following structure:

```json
{{
  "common": [
    {{"A_index": X, "B_index": Y}},
    ...
  ],
  "only_in_A": [X, ...],
  "only_in_B": [Y, ...]
}}
```

Where:
- `A_index` is the index of the item in List A,
- `B_index` is the index of the semantically equivalent item in List
↪  B,
- `only_in_A` lists the indices of items in A that do **not** appear
↪  in B,
- `only_in_B` lists the indices of items in B that do **not** appear
↪  in A.

# Notes
- Both List A and B are numbered using 1-based indexing.
- Match items even if they are paraphrased or formatted
↪  differently.
- Treat legal synonyms and abbreviations as equivalent when
↪  appropriate.
- Return only valid JSON in the **Final Answer** section.
```

Figure 28

```
┌─────────────────────────────────────────────────────────────────────┐
│ ███ Prompt for Extract Residual Facts from Uncovered Text by the Checklist Items ███ │
│                                                                         │
│   You are assisting a lawyer in identifying key information from a      │
│   ↪  legal case summary. You will be given a set of text spans         │
│   ↪  extracted from the summary that may contain meaningful legal or    │
│   ↪  factual content.                                                   │
│                                                                         │
│   Your task is to extract distinct atomic facts from the given spans.  │
│   ↪  Each atomic fact should be a single discrete, self-contained,      │
│   ↪  and verifiable piece of information that can stand on its own.     │
│   ↪  Ignore any spans that contain filler phrases, incomplete          │
│   ↪  clauses, or do not convey meaningful information. If multiple      │
│   ↪  spans express the same fact, extract it only once.                │
│                                                                         │
│   # Note: Do not make assumptions or add information that is not        │
│   ↪  present in the spans.                                              │
│                                                                         │
│   # Text Spans                                                          │
│   {text_spans}                                                          │
│                                                                         │
│   # Output Format                                                       │
│                                                                         │
│   Your output should be in the following JSON format-no extra keys,    │
│   ↪  no prose outside of the JSON:                                      │
│                                                                         │
│   ```                                                                   │
│   {{                                                                    │
│     "reasoning": "<brief analysis of which spans contain meaningful    │
│     ↪  factual information and what those facts are>",                  │
│     "extracted": [                                                      │
│       {{                                                                │
│         "fact": "<atomic fact 1>",                                      │
│         "evidence_spans": [<list of 1-based span indices>]             │
│       }},                                                               │
│       {{                                                                │
│         "fact": "<atomic fact 2>",                                      │
│         "evidence_spans": [<list of 1-based span indices>]             │
│       }}                                                                │
│       // ...                                                            │
│     ]                                                                   │
│   }}                                                                    │
│   ```                                                                   │
│                                                                         │
│   ## Definitions of each part                                          │
│   * `reasoning`: A brief analysis of the spans and how you identified  │
│   ↪  any meaningful atomic facts.                                       │
│   * `extracted`: A list of objects, each representing one atomic fact. │
│   ↪  Every object must have:                                           │
│     - `fact`: A clear, concise sentence or phrase conveying a          │
│     ↪  distinct, self-contained fact.                                   │
│     - `evidence_spans`: A list of 1-based indices of the spans that    │
│     ↪  support or directly contain the fact.                           │
│                                                                         │
│   ## Rules for the JSON schema                                         │
│   {it is the same as the checklist extraction prompt.}                 │
└─────────────────────────────────────────────────────────────────────┘
```

Figure 29

```
1998
1999
2000
2001
2002
2003
2004
2005
2006
2007
2008
2009
2010
2011
2012
2013
2014
2015
2016
2017
2018
2019
2020
2021
2022
2023
2024
2025
2026
2027
2028
2029
2030
2031
2032
2033
2034
2035
2036
2037
2038
2039
2040
2041
2042
2043
2044
2045
2046
2047
2048
2049
2050
2051
```

**Prompt for Rating Writing Style Similarity on Five Aspects**

```
You are given two summaries of the same legal case (Summary A and
↪   Summary B). Your task is to evaluate how similar they are in
↪   terms of structure and writing style across five specific
↪   dimensions. You should focus on **similarity** rather than
↪   quality-we want to know how alike these summaries are, not
↪   which one is better.

# Summaries to Compare
## Summary A:
{summary_A}

## Summary B:
{summary_B}

# Evaluation Dimensions with Specific Similarity Scales

{all_5_aspects_definitions}

# Output Format

Structure your response as follows:

**Analysis:**
Provide a detailed comparison for each dimension, explaining
↪   specific similarities and differences you observe between
↪   Summary A and Summary B.

**Scores:**
Output a valid JSON object with your similarity ratings:

```json
{{
  "readability_jargon": X,
  "narrative_order": X,
  "sentence_structure": X,
  "formatting_layout": X,
  "citation_style": X
}}
```

Where X is your similarity rating (1-5) for each dimension.

# Important Notes
- Focus on similarity, not quality or factual correctness
- Evaluate style and structure only, ignore content differences
- Consider the summaries as a whole when rating each dimension
- Apply the scale objectively for every dimension, strictly
↪   following each definition
```

Figure 30

Prompt for Legal Summarization

```
You are given multiple documents related to a legal case. Your task
↪  is to generate a clear, legally precise, and self-contained
↪  summary that would let the reader grasp the case without
↪  consulting the source files without being excessively long or
↪  overly detailed.

Write the summary as a factual narrative. The checklist below shows
↪  what to include. Items marked "(if applicable)" should only be
↪  included when relevant. If information isn't in the documents,
↪  omit it-do not speculate.

# Legal Case Summary Checklist
{all_26_checklist_item_definitions}

# Case Documents
{case_documents}

# Output Format
Please structure your response as follows:
**Reasoning:** Briefly explain what key elements you focused on in
↪  the documents to build your summary.

**Case Summary:** A clear, legally precise narrative of the case,
↪  written in paragraph form, without being too long.

# Guidelines
* Write as a narrative in paragraph form using clear language. Use
↪  a logical order-chronological if helpful, but flexible if
↪  another sequence improves clarity.
* Include enough detail for understanding while remaining concise.
* Use accurate legal terminology but avoid jargon-write for a
↪  general audience.
* Stay strictly factual; do not add analysis beyond what appears in
↪  the record.

Now read the case documents and generate the summary following the
↪  checklist, output format, and guidelines above.
```

Figure 31

```
Prompt for End-to-End Extracting Checklist Item from Case Document (Part 1/2)

You are assisting a lawyer in extracting key information from legal
↪  case documents. You will be given multiple documents related to
↪  a legal case. Your task is to {item_description}

# Note:
- Do not make assumptions or add information that is not presented
↪  in the documents.
- When extracting evidence, quote the exact text from the
↪  documents.
- Each extracted value must be self-contained and easy to
↪  understand; include important context when available.

# Case Documents
{case_documents}

# Output Format
Your output should be in the following JSON format-no extra keys,
↪  no prose outside of the JSON:

```
{
  "reasoning": "<brief analysis of the case documents and how you
  ↪  identified the relevant information or determined that none
  ↪  was present>",
  "extracted": [
    {
      "evidence": [
        {
          "text": "<verbatim snippet 1>",
          "source_document": "<document name>",
          "location": "<page number or section>"
        },
        {
          "text": "<verbatim snippet 2 (if multiple snippets are
          ↪  relevant)>",
          "source_document": "<document name>",
          "location": "<page number or section>"
        }
        // ...
      ],
      "value": "<extracted information from the evidence>"
    }
    // ...
  ]
}
```
```

Figure 32

2160
2161
2162
2163
2164
2165
2166
2167
2168
2169
2170
2171
2172
2173
2174
2175
2176
2177
2178
2179
2180
2181
2182
2183
2184
2185
2186
2187
2188
2189
2190
2191
2192
2193
2194
2195
2196
2197
2198
2199
2200
2201
2202
2203
2204
2205
2206
2207
2208
2209
2210
2211
2212
2213

```
Prompt for End-to-End Extracting Checklist Item from Case Document (Part 2/2)

## Definitions of each part
- `reasoning`: A brief analysis of the case documents and how you
↪  identified the relevant information or determined that none was
↪  present.
- `extracted`: A list of one or more objects, each representing a
↪  distinct piece of information relevant to the checklist item.
↪  Always use a list, even if there is only one item.
- `evidence`: A list of evidence objects, each containing:
  - `text`: Exact text snippet copied from the case documents
  - `source_document`: The title/name of the document where this
  ↪  evidence was found
  - `location`: The page number or section identifier where the
  ↪  evidence appears
- `value`: The extracted information based on the evidence.

## Rules for the JSON schema
1. **extracted** and **evidence** are always lists, even if they
↪  hold a single object.
2. Copy the **text** in evidence objects exactly as it appears in
↪  the case documents-no rewriting or paraphrasing.
3. Always include **source_document** and **location** for each
↪  piece of evidence.
4. If the case documents contain no relevant information, output
↪  the **extracted** as an empty list:

```
{
  "reasoning": "<brief analysis>",
  "extracted": []
}
```

5. Extract information from all relevant documents-do not stop
↪  after finding information in just one document.
6. Each distinct piece of information should be a separate item in
↪  the **extracted** list.
7. If you cannot determine the specific page number or section, you
↪  may use descriptive locations like "beginning of document",
↪  "middle section", or "near the end".
```

Figure 33

```
Prompt for Chunk-by-Chunk Extracting Checklist Items from Case Documents

You are assisting a lawyer in extracting key information from legal
↪  case documents. You will be given a document chunk from a legal
↪  case. Your task is to {item_description}

# Note:
{same as the end-to-end prompt}

# Current State
This is the accumulated extraction state from previous chunks:
{current_state}

# Document Information
- Document Name: {document_name}
- Chunk: {chunk_id}/{total_chunks}

# Document Chunk
{document_chunk}

# Output Format
Your output should be in the following JSON format-no extra keys,
↪  no prose outside of the JSON:

```
{{
  "reasoning": "<brief analysis of this document chunk and how you
  ↪  identified any new relevant information or determined that
  ↪  none was present>",
  "extracted": [
    {{
      "evidence": [
        {{
          "text": "<verbatim snippet 1>",
          "source_document": "<document name>",
          "location": "Chunk {chunk_id}/{total_chunks}"
        }},
        {{
          "text": "<verbatim snippet 2 (if multiple snippets are
          ↪  relevant)>",
          "source_document": "<document name>",
          "location": "Chunk {chunk_id}/{total_chunks}"
        }}
        // ...
      ],
      "value": "<extracted information from the evidence>"
    }}
    // ...
  ]
}}
```

## Definitions of each part
{same as the end-to-end prompt}

## Rules for the JSON schema
{{same as the end-to-end prompt}}
```

Figure 34

2268
2269
2270
2271
2272
2273
2274
2275
2276
2277
2278
2279
2280
2281
2282
2283
2284
2285
2286
2287
2288
2289
2290
2291
2292
2293
2294
2295
2296
2297
2298
2299
2300
2301
2302
2303
2304
2305
2306
2307
2308
2309
2310
2311
2312
2313
2314
2315
2316
2317
2318
2319
2320
2321

**System Prompt used in GAVEL-AGENT (Part 1/3)**

```
You are a document extraction specialist. Your task is to extract
↪  **all checklist items specified in the snapshot** from the
↪  provided documents, citing evidence for every value.

You operate by analyzing the snapshot and selecting **exactly ONE
↪  action per turn**. You must **respond with valid JSON only**
↪  - no prose, no extra keys.

# Snapshot
Provided every turn:
- Task description
- Checklist definitions (what items to extract; any number of
↪  items)
- Document catalog with coverage statistics (and
↪  catalog_state/version)
- Checklist summary (which keys are filled/empty/Not Applicable)
- Recent action history

# Goal
Systematically extract all applicable checklist items with proper
↪  evidence.

# Decision Policy
Choose exactly one action each turn:
- If the document catalog is **unknown** -> call `list_documents`.
- If a specific document likely contains a target value, choose
↪  ONE:
  * `read_document` - default choice. Read a targeted window
  ↪  (<=10,000 tokens) in a document.
  * `search_document_regex` - use this when the target is clearly
  ↪  patternable (e.g., "Case No.", "Filed:", citations).
- When you have confirmed text for one or more keys:
  - Use `append_checklist` for adds new entries for some checklist
  ↪  items.
  - Use `update_checklist` to replace the entire extracted list
  ↪  for some checklist items when you have the
  ↪  authoritative/complete set, when correcting earlier
  ↪  entries, or when setting an item to Not Applicable (see
  ↪  "Not Applicable Encoding").
- Periodically use `get_checklist` to assess remaining gaps.
- Stop when all keys are filled or set to Not Applicable.

# Systematic Extraction Process
**After each read_document or search_document_regex action:**
- Carefully analyze the returned text to identify ALL checklist
↪  items that can be extracted.
- Cross-reference the text against your checklist definitions to
↪  avoid missing relevant values.
- Your next action MUST be append_checklist or update_checklist
↪  if you found extractable values in the text just read.

**After each append_checklist or update_checklist action:**
- Verify whether all extractable values from the preceding text
↪  were included.
- If you notice missed values, immediately append them as the
↪  next action before continuing.
```

Figure 35

2322
2323
2324
2325
2326
2327
2328
2329
2330
2331
2332
2333
2334
2335
2336
2337
2338
2339
2340
2341
2342
2343
2344
2345
2346
2347
2348
2349
2350
2351
2352
2353
2354
2355
2356
2357
2358
2359
2360
2361
2362
2363
2364
2365
2366
2367
2368
2369
2370
2371
2372
2373
2374
2375

**System Prompt used in GAVEL-AGENT (Part 2/3)**

```
# Document Reading Efficiency
- **NEVER** reread fully visited documents (marked with  Fully
↪  Visited).
- **NEVER** reread token ranges already viewed (shown as "Viewed
↪  tokens: X-Y").
- When reading partially visited documents (marked with Partially
↪  Visited), read ONLY unviewed token ranges.
- Check the "Viewed tokens" list before calling read_document to
↪  avoid redundant reads.

# Write Semantics
- **Any checklist item can have multiple values**; the
↪  `extracted` field is always a list.
- **append_checklist**: add new entries; **Do not** set Not
↪  Applicable via `append_checklist`.
- **update_checklist**: replace the entire `extracted` list; use
↪  for single-valued items, complete/authoritative sets,
↪  corrections, or to set "Not Applicable".

# Evidence Requirements
- **Every extracted entry must include evidence** with:
  - `text` (verbatim snippet),
  - `source_document` (document name),
  - `location` (e.g., page, section, docket entry; include token
  ↪  offsets if available).

# Not Applicable Encoding
- Represent Not Applicable as a **single extracted entry** for
↪  that key, set **via `update_checklist`**:
  - `value`: **"Not Applicable"** (exact string; case-sensitive)
  - `evidence`: required (explicit text or a dispositive posture
  ↪  supporting Not Applicable)
- A key is treated as **Not Applicable** only if its `extracted`
↪  list contains **exactly one** entry whose `value` is "Not
↪  Applicable".
- Do **not** mark Not Applicable solely because you failed to
↪  find a value; require explicit text or logically dispositive
↪  evidence (e.g., dismissal with prejudice -> no
↪  settlement/decree; "no class certification sought" -> class
↪  action items Not Applicable).
- If later evidence shows the item **does** have real values, use
↪  `update_checklist` to replace the Not Applicable entry with
↪  the confirmed entries.

# Stop Criteria
- Stop only when every checklist key is either:
  * Complete: all relevant values present in the corpus for that
  ↪  key have been extracted, each with evidence.
  * Not Applicable: represented as a single extracted entry with
  ↪  value "Not Applicable" and supporting evidence.
- Before stopping, verify state with `get_checklist` (in a prior
↪  turn if needed) and, if consolidation is required, issue one
↪  final `update_checklist` (in a prior turn) to replace any
↪  incrementally built keys with their curated final lists. Then
↪  return the stop decision.
```

Figure 36

**System Prompt used in GAVEL-AGENT (Part 3/3)**

```
{{TOOL_DESCRIPTIONS}}

# Response Format
- On each assistant turn, do exactly **one** of:
  1) **Issue one function call**, or
  2) **Stop** if all applicable checklist items are fully
  ↪  extracted and any non-applicable items are marked.
- When stopping, return **only** this JSON (no extra text):
```json
{
  "decision": "stop",
  "reason": "<brief justification>"
}
```

Figure 37

