# OpenReview forum: "Gavel: Agent Meets Checklist for Evaluating LLMs on Long-Context Legal Summarization"
_ICLR.cc/2026/Conference — ICLR 2026 Conference Withdrawn Submission_

### Official Review · Reviewer_8q7c · 2025-10-31

**Soundness:** 2
**Presentation:** 2
**Contribution:** 2
**Rating:** 4
**Confidence:** 4

**Summary:**

The authors introduceGavel-Ref, a reference-based evaluation framework that aims to improve checklist evaluation on the long-context task of legal summarization. Their evaluation shows that frontier models succeed on simple checklist items but struggle on multi-value or rare ones such as settlements and monitor reports.

**Strengths:**

The authors defined a reference-based evaluation framework for comprehensively assessing legal summarization with checklist-style, residual facts, as well as writing style evaluations. This allows for more nuanced evaluation beyond a single-scaler score.

**Weaknesses:**

- The amount of the cases (50) used in the evaluation is very limited. It is also unclear how such cases are selected. Given so many different legal areas, types of legal documents, juridictions etc., the representativeness of the 50 cases is questionable. For instance, a "good" summary for a contract law case could look very differently to a criminal law case.

- Judgement from LLMs are used extensively in GAVEL-REF framework, yet it's unclear which model(s) are used in which step. There is no information as e.g. how self-bias and family-bias are taken into account in the evaluation.

- In section 2.3, the authors "recruit four in-house annotators with legal expertise", yet it remains unclear what level of expertise the annotators acquire and in which jurisdiction. The recruitment criteria and procedure is also not presented. Given the high-stake nature of legal domain, this presents a significant limitation.

- While GAVEL-Agent shows advantage over other methods such as end-to-end and chunk-by-chunk ones, and the tools defined in Section 4.1 may have great potentials as a product, I question the extent to which engineering efforts can translate into research insights.

**Questions:**

- Given the difficulty of legal summarization, it would be very helpful to include qualitative analyses on what kind of cases lead to lower vs. higher scores.

- In section 4.2, could you provide the rationale of the model selection?

---

> ### Author Response · Authors · 2025-12-01
> **Author Response (#1)**
>
> Thank you for your valuable feedback on our work. We appreciate the opportunity to address your comments. In light of feedback from you and the other reviewers, we have doubled the number of cases used in our paper: from 20 to 40 for meta-evaluation of Gavel-Ref and the document checklist extraction methods (Sections 2 and 4), and from 50 to 100 for legal summarization evaluation (Section 3). We have updated the paper with the new results.
>
> In this response, we address your specific points. For other changes made to the paper, please refer to the general response.
>
> > The amount of the cases (50) used in the evaluation is very limited. It is also unclear how such cases are selected. Given so many different legal areas, types of legal documents, juridictions etc., the representativeness of the 50 cases is questionable. For instance, a "good" summary for a contract law case could look very differently to a criminal law case.
>
> We agree that increasing the number of cases used to evaluate LLMs is important, so we have doubled the evaluation set from 50 to 100 cases, with 20 cases per length bin (32K, 64K, 128K, 256K, and 512K tokens). The 100 cases align with the scale of the legal summarization task in the previous benchmarks such as ExpertLongBench and Helmet (ICLR 2025). For case selection, as described at the beginning of Section 3, we prioritize minimizing data contamination: 83 out of 100 cases are filed in 2025. The remaining 17 cases are from earlier years due to limited availability in 2025—especially in the 512K bin. On average, cases in the 512K bin take about 1.5 years to accumulate that much text, while at the time of writing the paper, we only have ~7 months of 2025 data, so it is not feasible to fill this bin using only 2025 filings.
>
> Regarding legal areas, our focus on civil-rights cases is driven by the data source: the Civil Rights Litigation Clearinghouse, which provides both case documents and expert-written summaries specifically for civil-rights litigation. As a result, our study does not cover other domains such as contract or criminal law. We agree that extending this type of evaluation to additional areas of law is valuable, which we leave for future work.
>
> Within the civil rights area, we strive for diversity across case types, subject to our contamination-control and length-binning criteria. Across the 100 cases, we cover 19 distinct case types. The largest categories are Presidential Authority (34 cases) and Immigration and/or the Border (20), followed by Education (9), Speech and Religious Freedom (9), Environmental Justice (3), Equal Employment (3), Prison Conditions (3), and twelve additional case types with smaller counts. Thus, while our dataset is not intended to represent all legal domains, it does include a range of civil-rights case types under realistic long-context conditions.
>
>
> > Judgement from LLMs are used extensively in GAVEL-REF framework, yet it's unclear which model(s) are used in which step. There is no information as e.g. how self-bias and family-bias are taken into account in the evaluation.
>
> We actually listed the models used for evaluation on Lines 229-230: “We use GPT-oss 20B for checklist extraction and style rating, and Gemma3 27B for checklist comparison in Section 3 to evaluate LLM summaries.” These models are chosen based on the meta-evaluation in Table 1, where they are the best-performing open-source models for their respective tasks.
>
> Regarding self-bias and family-bias: unlike typical LLM-as-judge settings where a model directly “rates” the quality of its own (or related) outputs, our models mainly perform information extraction and matching:
> - For checklist extraction and checklist comparison, the tasks involve extracting structured fields and checking whether two texts contain the same information, rather than assigning subjective quality scores.
> - For style rating, the model evaluates the style similarity between two summaries not absolute style quality. Prior work such as [1] show such similarity based judgements exist no self-bias.
>
> [1] SimulatorArena: Are User Simulators Reliable Proxies for Multi-Turn Evaluation of AI Assistants? EMNLP 2025.

---

> ### Author Response · Authors · 2025-12-01
> **Author Response (#2)**
>
> > In section 2.3, the authors "recruit four in-house annotators with legal expertise", yet it remains unclear what level of expertise the annotators acquire and in which jurisdiction. The recruitment criteria and procedure is also not presented. Given the high-stake nature of legal domain, this presents a significant limitation.
>
> All four annotators are native English speakers and U.S.-based undergraduate students with basic familiarity with legal cases. We train them by carefully reviewing all legal terms used in the checklist (e.g., decree, settlement, ruling) and the relevant common background, and we walk through examples to ensure they clearly understand each checklist item. Their task is to read legal summaries, which are written for laypeople, and extract checklist items, not to read full case documents or write summaries themselves. Once the checklist definitions are clear, this annotation task does not require formal legal training.
>
> To avoid overstating their background, we have removed the phrase “with legal expertise” from the main text and added these recruitment and training details to Section D in the Appendix.
>
>
>
> > While GAVEL-Agent shows advantage over other methods such as end-to-end and chunk-by-chunk ones, and the tools defined in Section 4.1 may have great potentials as a product, I question the extent to which engineering efforts can translate into research insights.
>
> The main insight behind Gavel-Agent is to propose and systematically study an agent-based approach for long-context evaluation. Prior work has largely relied on two standard strategies: (i) end-to-end prompting, where all text is fed into a long-context model, and (ii) chunk-by-chunk extraction. Our results show that, as LLMs improve in tool use and reasoning, an agent scaffold can offer a better efficiency–quality trade-off than these baselines. In our updated meta-evaluation on checklist extraction from case documents, Gavel-Agent with Qwen3 30B-A3B (26-agent setup) achieves the second-best performance—only 7% lower than the GPT-4.1 end-to-end method—while reducing token usage by 36% (and 59% relative to the chunk-by-chunk variant with the same backbone).
>
> At the same time, there remains substantial headroom for future work: our best Gavel-Agent configuration reaches an $S_\text{checklist}$ of 43.5 out of 100, far below what is achieved when extracting from summaries. We also find that current open-source LLMs do not handle many checklist items at once very well: grouped and single-agent setups save tokens but yield worse performance than the 26-agent configuration, revealing a clear trade-off between the number of agents and the number of items each agent handles. More advanced agent designs (e.g., multi-agent coordination, improved planning, or RL-based training) could further improve long-context evaluation.
>
> Overall, we demonstrate the practical promise of the agent-based methods for long-context evaluation and identify clear gaps that future research can improve.
>
>
> > Given the difficulty of legal summarization, it would be very helpful to include qualitative analyses on what kind of cases lead to lower vs. higher scores.
>
> With the updated evaluation on 100 cases, we find that cases on the longer side (256K and 512K bins) consistently lead to lower scores for all 12 LLMs (see Figure 2).
>
> > In section 4.2, could you provide the rationale of the model selection?
>
> In Section 4.2, our goal is to meta-evaluate different checklist extraction methods rather than to exhaustively benchmark models, so we select models that are representative of the constraints of each setting.
>
> For end-to-end extraction, we use GPT-4.1 because it is one of the few models that natively supports a 1M-token context window given we want to feed all case documents at once. For chunk-by-chunk extraction and Gavel-Agent, efficiency and cost become more important since the model is invoked many times and we evaluate multiple scaffold configurations. We therefore focus on the best performing open-source reasoning models, GPT-oss 20B, Qwen3 32B, and Qwen3 30B-A3B. For the agent setting, we need models with a long native context window (≥128K tokens) to include the action history; Qwen3 32B only supports 32K context natively, so we use Qwen3 30B-A3B and GPT-oss 20B for our Gavel-Agent scaffold.

---

### Official Review · Reviewer_bwrh · 2025-11-01

**Soundness:** 2
**Presentation:** 4
**Contribution:** 2
**Rating:** 4
**Confidence:** 4

**Summary:**

This paper studies content selection and stylistic evaluation of text summaries generated from long legal documents. They combinechecklist-based evaluation with residual facts and writing style metrics. First, the authors evaluate a suite of open-source and proprietary LLMs on Gavel-Ref dataset. Second, they explored methods to automatically extract checklist answers from source documents.

Gavel-Ref is an extension of a prior benchmark (ExpertLongBench). ExpertLongBench includes a pre-defined set of 26 checklist items that are important for legal document summarization. Each input is already annotated with reference summary. Gavel-Ref makes three key extensions to ExpertLongBench. It modifies checklist answers to include multiple values. Additionally, Gavel-Ref includes atomic facts not covered by the checklists (residual facts) and an evaluation of the writing style on a Likert scale. The paper also includes a meta-evaluation of their LLM-based extraction of checklist answers from system-generated summaries and reference summaries. The paper evaluates a suite of 12 open-source and proprietary LLMs on Gavel-Ref. A notable result here is that the most LLMs often achieve higher scores at longer input lengths (>=128k) than shorter inputs (<64k) (Figure 2). This is a stark difference from prior evaluations on long-context benchmarks.

The second part of the paper explores three methods to automatically extract answers to the checklist questions from the source documents: end-to-end long-context model, chunk-by-chunk and a agent-based system. They evaluate these three method against reference-based checklist from Gavel-Ref. GPT-4.1 based long-context model performs the best, while the agent-based system (Qwen3-30B-A3B, gpt-oss-20B) significantly underperforms.

Overall, the paper studies an important evaluation task but I have some concerns with the results (at varying context lengths) and agent-related experiments.

**Strengths:**

- Checklist-based evaluation is a reliable option for certain technical domains such as finance and legal. This paper highlights key limitations with an existing checklist-based benchmark (ExpertLongBench) and proposes fixes through multi-value answers and residual facts.
- The paper also explores the idea of extracting answers to the checklist directly from source documents. This is especially important for long input tasks because human written checklists are expensive to collect (if feasible). While I think an LLM agent is a strong option for this task, the results show that the agent approach underperforms a long-context model.
- The context management component of Gavel-Agent is quite interesting, and well suited for the task.

**Weaknesses:**

- The paper needs additional discussion of the results from Figure 2. Its unclear (and a bit surprising) why the models perform better at longer inputs than shorter inputs. This seems counter-intuitive and some qualitative analysis here could be helpful.
- The proposed agent-based method significantly underperforms a strong long-context model (GPT-4.1). To show that the agent approach is reliable, it would be interesting to explore stronger models within the agent setup. I understand the argument around efficiency and performance, but for extracting reference checklists I think performance is very critical.
- The paper doesn't discuss the effect of summary length. Longer system-generated summaries have a higher chance of including both checklist and residual facts. Do you control for summary length across the evaluated systems?

**Questions:**

Some additional questions and comments on the paper,

- Gemini Flash outperforms Pro and Claude Sonnet outperforms Opus on Gavel-Ref – this is a bit unusual. I don't think I fully agree with the long-context argument provided in lines 252-254. Prior benchmarks shows that Gemini Pro has a longer effective context window length. Additional analysis here could be really helpful.
- Is the size of checklist fixed to 26 items? Even when extracting they are automatically extracted from the source documents?
- Human-curated checklists could be expensive (or even infeasible) for longer documents, so it would be interesting to check if automatically extracted (end-to-end or agent-based) checklist values improve on recall compared to human-curated checklist. Maybe a human expert could verify the automatically extracted checklists?
- I am not sure I fully agree with the drawback of ExpertLongBench from lines 79-81. I believe it is possible for the user to breakdown the performance by task and example.

---

> ### Author Response · Authors · 2025-12-01
> **Author Response (#1)**
>
> Thank you for your valuable feedback on our work. We appreciate the opportunity to address your comments. In light of feedback from you and the other reviewers, we have doubled the number of cases used in our paper: from 20 to 40 for meta-evaluation of Gavel-Ref and the document checklist extraction methods (Sections 2 and 4), and from 50 to 100 for legal summarization evaluation (Section 3). We also refined the definitions of the checklist items after reviewing both human annotations and model-extracted checklists and have updated the paper with the new results, which is more accurate. As a result, we substantially rewrote Section 3, and several earlier findings (including some you questioned) have changed.
>
> Below we summarize the updated results most directly related to your comments; additional analyses can be found in Section 3 and Section E of the Appendix. For other changes made to the paper, please refer to the general response.
>
> > Gemini Flash outperforms Pro and Claude Sonnet outperforms Opus on Gavel-Ref – this is a bit unusual.
>
> In the updated results, Gemini 2.5 Pro now achieves the best overall $S_\text{Gavel-Ref}$ among all models, outperforming Gemini 2.5 Flash. Within the Claude family, Sonnet 4 still outperforms Opus 4.1. To better understand this gap, we provide checklist item–level performance for each LLM in Figures 10–12 in the Appendix. We find that Sonnet 4 is stronger than Opus 4.1 on several key items, such as _Cause of action_, _Class action vs. individual_, and _Remedy sought_.
>
>
> > It’s unclear (and a bit surprising) why the models perform better at longer inputs than shorter inputs.
>
> In the updated results, this pattern no longer holds. We now observe a consistent trend across all models: performance _decreases_ as case length increases, with models performing worst on the 256K and 512K bins. For example, Gemini 2.5 Pro’s $S_\text{Gavel-Ref}$ score on 512K cases is 4.7 points lower than on 32K cases, and GPT-4.1 drops by 7.6 points between these bins.
>
>
> > Is the size of the checklist fixed to 26 items? Even when extracting they are automatically extracted from the source documents?
>
>
> Yes, the checklist is always fixed to 26 items, even for automatic extraction from source documents.
>
> However, not all items are applicable to every case. In the 40 human-annotated cases, a summary contains on average 16.3 out of 26 items. For both summary-based and document-based extraction, the model simply leaves an item empty if it is not applicable or if it cannot find evidence for it.
>
>
> > Human-curated checklists could be expensive (or even infeasible) for longer documents, so it would be interesting to check if automatically extracted (end-to-end or agent-based) checklist values improve on recall compared to human-curated checklist
>
> As shown in Figure 5, the best-performing method (end-to-end extraction with GPT-4.1) and the second-best (Gavel-Agent with Qwen3 30B-A3B) achieve $S_\text{checklist}$ scores of 46.9 and 43.5, respectively. This indicates that current document-level extraction methods are still far from reliable enough to recover all relevant checklist information from the case documents.
>
> To make this more concrete, we also include a side-by-side visualization comparing the checklist items extracted by Gavel-Agent from the case documents with those extracted by human annotators from the case summary. In most cases, the human-extracted checklists contain substantially more detail—especially for items such as Important Filings and Factual Basis. Since these checklists are derived from expert-written summaries, this suggests that human summarizers are currently much better at identifying and aggregating the key checklist information than automatic document-level extraction methods. Future work could explore human-in-the-loop approaches to study how current document-extraction methods support legal experts in writing summaries.

---

> ### Author Response · Authors · 2025-12-01
> **Author Response (#2)**
>
> > The paper doesn't discuss the effect of summary length. Longer system-generated summaries have a higher chance of including both checklist and residual facts. Do you control for summary length across the evaluated systems?
>
> The new Figure 6 shows a heatmap of each LLM’s average summary length in each case-length bin, alongside its overall $S_\text{Gavel-Ref}$ score. We find that proprietary models generally write longer summaries (>400 words) than open-source models (<400 words), while human summaries average 884 words. LLM-generated summaries are closer to human length in the 32K–128K bins, but for 256K and 512K cases, models produce much shorter summaries: human summaries are over 1,100 words, whereas LLMs are typically around 500–800 words. We also observe that GPT-5 tends to produce very long summaries (often >900 words) even for shorter cases, because it often writes in a list-style format rather than a concise narrative. Figure 9 shows a typical example.
>
> We further compute the correlation between summary length and $S_\text{Gavel-Ref}$. Overall, there is a moderate positive correlation (Pearson $r=0.31$, Spearman $\rho=0.36$, Kendall’s $\tau=0.24$), but this is largely driven by weaker open-source models that both underperform and produce shorter summaries.
> When we separate proprietary and open-source models, the correlations become much smaller: within proprietary models, Pearson $r=-0.11$, Spearman $\rho=-0.13$, and Kendall’s $\tau=0.09$; within open-source models, Pearson $r=0.20$, Spearman $\rho=0.20$, and Kendall’s $\tau=0.14$. This suggests that, once we control for model family, summary length alone explains only a small fraction of the performance differences.
>
>
>
> > I am not sure I fully agree with the drawback of ExpertLongBench from lines 79-81. I believe it is possible for the user to break down the performance by task and example.
>
> We agree that, in principle, users of ExpertLongBench could further break down performance by task and example. Our point was not that such analysis is impossible, but that the paper itself reports only a single aggregate number per model for the legal summarization task and does not provide item-level insights—for example, which checklist items models systematically struggle with or how well they capture non-checklist information. One of our contributions in Section 3 is to carry out exactly this kind of detailed analysis for legal summarization.
>
> We also agree that our original wording on lines 79–81 was too strong and could be read as overly critical of ExpertLongBench and related work, which make important contributions in introducing long-context benchmarks. Thus, we have revised the text to:
>
> "Furthermore, ExpertLongBench and other existing benchmarks (Yen et al., 2024; Ruan et al., 2025)are built to evaluate LLMs across many tasks, with legal summarization as one of them. They provide valuable benchmarking of these models, but they naturally do not aim to offer detailed analysis of how modern LLMs perform on legal summarization specifically"

---

### Official Review · Reviewer_W5hh · 2025-11-01

**Soundness:** 2
**Presentation:** 2
**Contribution:** 1
**Rating:** 4
**Confidence:** 3

**Summary:**

This paper develops a reference-based evaluation framework for assessing legal summarization. It follows up on a checklist comparison approach to evaluating legal summaries and also proposes a new agent scaffold to assist the extraction of the checklist items. Results show that GPT 4.1 performs the best in the end-to-end scenario under a long context.

**Strengths:**

I love that the evaluations are very comprehensive, spanning across 12 frontier models. In addition, I appreciate your efforts in conducting a relatively large scale human annotation effort, which enhances the rigor of this study. There are also in-depth analyses of the failure modes and how top models succeed in this task, which gives us more insights.

**Weaknesses:**

My biggest issue with the paper is that the contribution feels very incremental. The main method is a follow-up on an existing paper (https://arxiv.org/pdf/2506.01241) which is not a popular and widespread evaluation method as of today. Also, the existing method from Ruan et al. (2025) relied on 26 items from the legal experts in the study, and it really feels like the premise of Ruan et al. (2025) needs to be more general and robust. In addition, this paper presents a low inter-annotator agreement, which weakens the validity of evaluation.

**Questions:**

Suggestions:
1. Figure 2 does not have to go all green. You can go with a wider gradient of color given that for each task the colors in the heatmap are a bit indistinguishable. Widening the spectrum will help readers differentiate.
2. Figure 3 feels a bit a convoluted and would be nice if a qualitative example or a clearer message is shown.

---

> ### Author Response · Authors · 2025-11-22
> **Author Response (#1)**
>
> Thank you for your valuable feedback on our work. We appreciate the opportunity to address your comments.
>
> > My biggest issue with the paper is that the contribution feels very incremental. The main method is a follow-up on an existing paper (https://arxiv.org/pdf/2506.01241) which is not a popular and widespread evaluation method as of today. Also, the existing method from Ruan et al. (2025) relied on 26 items from the legal experts in the study, and it really feels like the premise of Ruan et al. (2025) needs to be more general and robust.
>
> We respectfully disagree with the assessment that our contribution is incremental. The focus of our paper is on evaluating legal summarization. As we describe in Introduction (Line 052-053) and Related Work (Lines 448-458), checklist-based evaluation has become a popular method in the era of LLM-as-judge. In this context, ExpertLongBench (Ruan et al., 2025) is a very recent and highly relevant work: legal summarization is one of their 13 long-context tasks, and they curate a set of 26 checklist items with legal experts. One of their co-authors, Margo Schlanger, is also the founder and director of the Civil Rights Litigation Clearinghouse, which is the source of our cases and reference summaries. Thus, ExpertLongBench provides a natural starting point for building a rigorous evaluation of legal summarization, rather than an isolated or “niche” method.
>
> Our work goes well beyond simply adopting their checklist. Compared to ExpertLongBench and other related work, we make three main contributions (as summarized in the Introduction):
> 1. __Gavel-Ref: an improved checklist-based evaluation framework.__ We extend the ExpertLongBench checklist evaluation in several ways: we support multi-value extraction with associated supporting text, list-wise comparison rather than single-value comparison only, and a principled way of score aggregation. In addition, we introduce two complementary components that ExpertLongBench does not cover: residual facts (information beyond the 26 expert items) and writing-style evaluation. Together, these compose Gavel-Ref, a more comprehensive reference-based evaluation framework for legal summarization (Section 2).
> 2. __A systematic, fine-grained evaluation of 12 frontier LLMs.__ Using GAVEL-REF, we conduct a detailed evaluation of 12 state-of-the-art models on legal summarization. Beyond overall scores, we report results separately for checklist, residual facts, and style (Section 3.1), analyze performance by checklist groups (Section 3.2), and perform checklist item–level analysis, including top/bottom items and over-/under-specified items (Section 3.3). In contrast, ExpertLongBench reports only a single column of numbers for the legal summarization task in their Table 2, without performance breakdown or detailed analysis.
> 3. __Checklist extraction directly from case documents with an agentic scaffold.__ Beyond reference-based evaluation, we investigate a novel direction: automatically extracting checklist items directly from the underlying case documents, which enables scalable evaluation and testing of potentially superhuman models. We study three approaches–end-to-end extraction, chunk-by-chunk extraction, and our designed autonomous agent scaffold (Gavel-Agent)–which are not explored in ExpertLongBench. (Section 4)
>
> To make these contributions explicit in the paper, we will add the following text at the end of the Introduction:
> ```
> In summary, our contributions are as follows:
> 1. We introduce Gavel-Ref, a reference-based evaluation framework for legal summarization that provides comprehensive assessment via checklist, residual fact, and writing style evaluation.
> 2. Using Gavel-Ref, we systematically evaluate 12 frontier LLMs across different case lengths and reveal their gaps in capturing complex legal checklist items with detailed analysis.
> 3. We explore checklist extraction from case documents using three different approaches: end-to-end, chunk-by-chunk, and Gavel-Agent–our autonomous agent scaffold.
> ```

---

> ### Author Response · Authors · 2025-11-22
> **Author Response (#2)**
>
> > This paper presents a low inter-annotator agreement, which weakens the validity of evaluation.
>
> We ask human annotators to perform the same three tasks as the LLMs: checklist extraction, checklist comparison, and writing style rating. As reported in Section D (Lines 772-776), both checklist extraction and checklist comparison achieve high inter-annotator agreement (87.8 $S_\text{checklist}$ across annotators for checklist extraction, Fleiss’ $\kappa$ of 0.57 for single value checklist comparison, and an average pairwise F1 of 0.82 for multi-value checklist comparison). The low agreement (measured in Krippendorff’s $\alpha$ = 0.32) is on the writing style rating task, where annotators compare the style similarity between two summaries in 1-5 Likert scale along five aspects.
>
> This lower $\alpha$ reflects, to some extent, the inherent subjectivity of style judgments rather than widespread disagreement. We examine this further with the “two agree” metric and find that overall 94.4% of times, at least 2 annotators agree with each other on the rating, only 5.6% of times all three annotators choose different ratings. This indicates that most instances show clear majority agreement, and full disagreement is rare.
>
> ---
>
> Also, in light of all the feedback, we are doubling the number of cases used in our paper: from 20 to 40 for meta-evaluation of Gavel-Ref and the document checklist extraction methods (Sections 2 and 4), and from 50 to 100 for legal summarization evaluation (Section 3). We will update the manuscript with the new results and incorporate your feedback on the color scheme in Figure 2 and a qualitative example for Figure 3 within a few days. Please let us know if you have any other comments.

---

> ### Author Response · Authors · 2025-12-01
> **Author Response (#3)**
>
> Thank you again for your helpful feedback. We have now updated the paper with the new results; please see the general response and the revised manuscript for the changes.

---

### Official Review · Reviewer_U3Kr · 2025-11-01

**Soundness:** 2
**Presentation:** 3
**Contribution:** 2
**Rating:** 4
**Confidence:** 3

**Summary:**

The paper proposes GAVEL-REF to automatically evaluate LLM-generated summaries of legal cases. The paper focuses on long-context summarization, which aligns with the nature of legal documents. The paper reports results on several LLMs, such that GPT-4.1 performs the best while Qwen3 reduces token usage on 20 long case summaries.

**Strengths:**

The paper conducts a systematic evaluation of 12 frontier LLMs across five context length scales (32K–512K tokens), using predominantly 2025 cases (90% from 2025). This design offers credible insights into how state-of-the-art models process truly long contexts.

Instead of reporting only aggregate performance scores, the study provides fine-grained, item-level analyses, revealing that top-performing models achieve near-perfect accuracy on simple single-value items (e.g., filing date: 0.99).

The proposed GAVEL-AGENT introduces an innovative long-context extraction strategy that emulates how human experts selectively navigate documents. Its demonstrated 40–60% reduction in token usage—while maintaining competitive performance (31.9 vs. 43.7 S_checklist)—addresses critical real-world challenges of computational efficiency and cost.

The integration of three complementary evaluation dimensions—checklist accuracy, residual facts assessment, and writing style analysis—provides a more holistic view of summary quality compared to content-only metrics. Notably, the finding that GPT-5 retains more residual facts but exhibits weaker narrative structuring, while Claude models excel in stylistic fidelity, highlights the strength of this multi-dimensional evaluation framework.

**Weaknesses:**

The meta-evaluation in Section 2.3 is based on only 20 long summaries for checklist validation, with 15 receiving single annotations. Such a limited sample may fail to capture the full spectrum of edge cases and annotation disagreements across the 50 diverse cases evaluated in the main study.

The relatively low inter-annotator agreement for certain tasks (e.g., Krippendorff’s $\alpha$ = 0.32 for style ratings) questions the reliability of these annotations as ground-truth references.

Although the study aims to minimize data contamination by using mostly 2025 cases, it still includes five pre-2024 cases in the 512K-token bin due to “limited availability.” This exception weakens the overall contamination-control claim.

The GAVEL-REF metric (Equation 2) employs fixed hyperparameters ($\alpha$ = 0.9) without justification or sensitivity testing. While the dynamic weighting between checklist and residual facts based on proportion r appears intuitively reasonable, the lack of empirical validation or ablation studies leaves uncertainty about whether this configuration truly captures optimal summary quality or affects model rankings. This also precludes reproducibility.

Although the authors claim that GAVEL-AGENT is “fully customizable” for other domains, all experiments, evaluations, and validations are confined to U.S. civil rights cases from a single dataset (CRLC). The 26 checklist items are highly domain-specific, and the study offers no evidence that the framework, metrics, or agent scaffolding would generalize effectively to other legal systems, document types, or domains such as biomedical or financial texts.

**Questions:**

Why is the evaluation on 20 documents only? Is this a statistically large sample to convey any meaningful insights?

Why are the inter-annotator agreements on the lower side? Is the task complex/subjective? Does it not question the evaluation correctness?

Why is $\alpha$ set to 0.9?

---

> ### Author Response · Authors · 2025-11-22
> **Author Response (#1)**
>
> Thank you for your valuable feedback on our work. We appreciate the opportunity to address your comments. In light of feedback from you and the other reviewers, we are doubling the number of cases used in our paper: from 20 to 40 for meta-evaluation of Gavel-Ref and the document checklist extraction methods (Sections 2 and 4), and from 50 to 100 for legal summarization evaluation (Section 3). In this response, we clarify some of your points. We will update the manuscript with the new results and post another response that addresses your remaining comments (and any new ones) in a few days.
>
> > The meta-evaluation in Section 2.3 is based on only 20 long summaries for checklist validation, with 15 receiving single annotations. Such a limited sample may fail to capture the full spectrum of edge cases and annotation disagreements across the 50 diverse cases evaluated in the main study. Is 20 a statistically large sample to convey any meaningful insights?
>
> We appreciate the suggestion to increase the annotations for checklist extraction. First, we want to emphasize the human annotation efforts required to extract checklists from these 20 long summaries which is a non-trivial task. As described in Lines 190-198, an annotator must extract all values for 26 checklist items per case, which takes about one hour per summary. In total, we collect 2,934 item-level annotations (about 40 annotation hours with \$800). To illustrate the density and granularity of the annotations for each case, we are going to include a full example of one such human-annotated checklist alongside the values extracted by GavelAgent from the case documents in the Appendix.
>
> Besides, given the high annotation cost for this complex task, we deliberately focused on long summaries in the meta-evaluation to stress-test the extraction model. Longer summaries contain more events and information than shorter ones, so they are a harder setting for checklist extraction. The 20 long case summaries for meta-evaluation have an average length of 1,093 words (Line 192), while the case summaries for the 50 cases used for evaluating model summaries average 896 words (about 200 words shorter). Our rationale is that if the LLM can accurately extract checklist items from these longer summaries, it should perform at least as well on the somewhat shorter summaries used in the model evaluation.
>
> In response to the suggestion to enlarge the annotation set, we have invested an additional \$1000 annotation budget to extract checklists from 20 more summaries, with an average length of 1,167 words. We are running experiments and will update the new meta-evaluation results in a few days.
>
> > The relatively low inter-annotator agreement for certain tasks (e.g., Krippendorff's  = 0.32 for style ratings) questions the reliability of these annotations as ground-truth references. Is the task complex/subjective?
>
> We ask human annotators to perform the same three tasks as the LLMs: checklist extraction, checklist comparison, and writing style rating. As reported in Section D (Lines 772-776), both checklist extraction and checklist comparison achieve high inter-annotator agreement (87.8 $S_\text{checklist}$ across annotators for checklist extraction, Fleiss'$\kappa$ of 0.57 for single value checklist comparison, and an average pairwise F1 of 0.82 for multi-value checklist comparison). The low agreement (measured in Krippendorff's $\alpha$ = 0.32) is on the writing style rating task, where annotators compare the style similarity between two summaries in 1-5 Likert scale along five aspects.
>
> This lower $\alpha$ reflects, to some extent, the inherent subjectivity of style judgments rather than widespread disagreement. We examine this further with the "two agree" metric and find that overall 94.4% of times, at least 2 annotators agree with each other on the rating, only 5.6% of times all three annotators choose different ratings. This indicates that most instances show clear majority agreement, and full disagreement is rare.

---

> ### Author Response · Authors · 2025-11-22
> **Author Response (#2)**
>
> > Although the study aims to minimize data contamination by using mostly 2025 cases, it still includes five pre-2024 cases in the 512K-token bin due to “limited availability.” This exception weakens the overall contamination-control claim.
>
> In total, we collect 50 cases binned by the total length of the case documents with 10 cases in each of 5 bins: 32K, 64K, 128K, 256K and 512K tokens. Of these, 45 are filed in 2025; the only exceptions are 5 cases in the 512K bin that are filed before 2025. This exception is purely due to data availability: in 2025 there are only 5 cases whose accumulated document length reaches the 512K bin. To fill this longest-length bin, we therefore have to source additional cases filed before 2025.
>
> Note that we use the filing date (the date of the first docket entry) as the case date. Our submission is in September 2025, so 2025 cases have at most about 6-7 months to accumulate documents. Reaching the 512K-token bin corresponds to roughly 384K words (around 1,400 book pages), which typically requires a long-running case. Indeed, for the 512K bin, the median duration between filing and the last document in our collection is 517 days (about 1.5 years), far longer than the 6-7 months available for 2025 cases. Therefore it is not feasible to fill the 512K bin with only 2025 filing cases, and we use 5 pre-2025 cases there while keeping the remaining 45/50 cases in 2025 to maintain our contamination-control goal.
>
>
> > The Gavel-Ref metric (Equation 2) employs fixed hyperparameters ($\alpha$= 0.9) without justification or sensitivity testing. While the dynamic weighting between checklist and residual facts based on proportion r appears intuitively reasonable, the lack of empirical validation or ablation studies leaves uncertainty about whether this configuration truly captures optimal summary quality or affects model rankings. This also precludes reproducibility. Why is $\alpha$  set to 0.9?
>
> In the evaluation literature, it is very common to manually set weights or coefficients across dimensions/rubrics when aggregating into an overall score. For example, MQM [1] (a widely used fine-grained human evaluation framework for machine translation) and HealthBench [2] (a recent checklist-based evaluation framework developed by OpenAI) both adopt manually set coefficients. In MQM, the final score is
>
> $$P = (\text{Issues}\_\text{minor} + \text{Issues}\_\text{major} \times 5 + \text{Issues}\_\text{critical} \times 10) / \text{Word count}$$
>
> where multipliers 5 and 10 are manually chosen to encode the relative severity of different error types.
>
> In our paper, the intuition behind the overall Gaver-Ref score is to prioritize content correctness, captured by the checklist and residual facts, over writing style, which we view as relatively easy to repair (e.g., by asking an LLM to rewrite a factually correct summary in a desired style). For this reason, we set $\alpha = 0.9$ in
>
> $$S_\text{Gavel-Ref} = (1 − r) · \alpha · S_\text{checklist} + r · \alpha · S_\text{residual} + (1 − \alpha) · S_\text{style}$$
>
> so that 90% of the weight goes to factual dimensions (checklist + residual) and 10% to style. The dynamic factor $r$ adjusts the relative importance of checklist versus residual facts based on their text proportion in a summary.
>
> Besides, when we present evaluation results for LLM-generated summaries in Section 3, we report all three component scores $S_{\text{checklist}}$, $S_{\text{residual}}$, $S_{\text{style}}$ alongside the overall Gavel-Ref score (see Figure 2). This allows readers to see how models behave on each dimension and, if desired, to reweight these components according to their own preferences. Because $\alpha$ and the full formula are explicitly specified, the metric is fully reproducible.
>
> [1] Multidimensional Quality Metrics: A Flexible System for Assessing Translation Quality, TL 2013
>
> [2] HealthBench: Evaluating Large Language Models Towards Improved Human Health, OpenAI 2025
>
> ---
> Please let us know if you have any other comments as we update the results.

---

> ### Author Response · Authors · 2025-12-01
> **Author Response (#3)**
>
> Thank you again for your helpful feedback. We have now updated the paper with the new results; please see the general response and the revised manuscript for the changes.

---

### Author Response · Authors · 2025-12-01
**General Response**

We thank all reviewers again for your valuable and thoughtful feedback. Based on your comments, we have made substantial changes to the paper. All edits are marked in blue in the updated paper. Below we summarize the main improvements and how they affect the results.

__1. Improved checklist definitions and doubled human checklist extraction annotations (20 -> 40 summaries)__

We improved 13 out of 26 checklist item definitions and doubled the number of human-annotated case summaries from 20 to 40. These annotations are used for:
- meta-evaluating checklist extraction from summaries (Section 2)
- meta-evaluating checklist extraction directly from case documents (Section 4)

After paper submission, we carefully reviewed both human annotations and model-extracted checklists and found some unclear or inconsistently interpreted items (e.g., distinguishing settlement vs. decree, where a settlement is what the parties privately agree, and a decree is a court-ordered settlement). We have updated the definitions of 13 items accordingly; the revised definitions are now displayed in Appendix B.

As described in Lines 206–215, these human-annotated checklists are costly and time-consuming: in total, we collect 5,442 item-level annotations at a cost of about $2,000 USD. Figures 13–22 provide an illustrative example, so readers can see the scale and granularity of the task. Given this cost, we decided to extend to 40 annotated summaries rather than more.

The main conclusions of the meta-evaluation still hold: GPT-5 remains the best model and GPT-oss 20B remains the second-best for checklist extraction from summaries.

__2. Fixing Gavel-Agent evaluation and updated findings__

After submission, we discovered an issue in the evaluation of Gavel-Agent: empty checklist items were not being handled correctly, which caused its reported performance to be lower than it should be. We fixed this issue and re-ran the document-level meta-evaluation on all 40 cases.

With the corrected evaluation, __Gavel-Agent with Qwen3 30B-A3B (26 individual agents)__ now achieves the second-best $S_\text{checklist}$, closely behind the end-to-end GPT-4.1 method. It is only about **7% lower in $S_\text{checklist}$** than GPT-4.1 end-to-end, while __reducing token usage by 36%__ (and by 59% compared to chunk-by-chunk with the same backbone).

This __reverses the earlier conclusion__ that Gavel-Agent performed worst and instead shows strong potential for agent-based approaches in long-context evaluation

__3. Doubling the legal summarization evaluation set (50 -> 100 cases) and revising Section 3__

We doubled the legal summarization evaluation set in Section 3 from 50 to 100 cases, with 20 cases per length bin (32K, 64K, 128K, 256K, and 512K tokens). The 100 cases align with the scale of the legal summarization task in the previous benchmarks such as ExpertLongBench and Helmet (ICLR 2025). With the refined checklist definitions, many of the findings in Section 3 changed, and we have __substantially rewritten Section 3__. Below we highlight the main differences and new analysis we added:
- Gemini 2.5 pro is now  the best model in terms of $S_\{Gavel-Ref}$ among the 12 LLMs, which was Gemini 2.5 Flash before.
- We now see a consistent pattern: __all models degrade as case length increases__, with performance lowest on 256K and 512K cases. In the original version, some models appeared to do better on longer cases.
- We added __a summary-length analysis__ comparing LLM-generated summaries with human summaries. We find that GPT-5 is an outlier, often producing very long summaries (~1,000 words) even on shorter cases where human summaries are around 500–700 words. We added __Figure 6__,  a heatmap of summary length vs. case length, and __Figure 9__, a qualitative example comparing GPT-5, Gemini 2.5 Pro, and human summary.
- We added __per-item performance plots__ for each of the 12 LLMs in Section E (Figures 10-12), which clarify which checklist items each model handles well or struggles with.

__4. Other writing and presentation improvements__

Besides these changes, we also revised the writing and presentation for clarity:
1. We added __a list of contributions__ at the end of the introduction.
2. We __rewrote the original Lines 79-81 (now Lines 80-83)__ to better acknowledge prior benchmarks such as ExpertLongBench and Helmet. The new text is:

"Furthermore, ExpertLongBench and other existing benchmarks (Yen et al., 2024; Ruan et al., 2025) are built to evaluate LLMs across many tasks, with legal summarization as one of them. They provide valuable benchmarking of these models, but they naturally do not aim to offer detailed analysis of how modern LLMs perform on legal summarization specifically"

We want to emphasize that we __greatly appreciate these works__ and intend our benchmark to be complementary, not critical.

3. In Figure 2 (benchmarking results for the 12 LLMs), we now __bold the best model in each column__ to make comparisons easier.

---

### Note · Authors · 2026-01-05

I have read and agree with the venue's withdrawal policy on behalf of myself and my co-authors.